# Revisiting Adversarial Training for ImageNet: Architectures, Training and Generalization across Threat Models

**Naman D Singh**[*]
University of Tübingen
Tübingen AI Center

**Francesco Croce**[*]
University of Tübingen
Tübingen AI Center

**Matthias Hein**
University of Tübingen
Tübingen AI Center

## Abstract

While adversarial training has been extensively studied for ResNet architectures and low resolution datasets like CIFAR-10, much less is known for ImageNet. Given the recent debate about whether transformers are more robust than convnets, we revisit adversarial training on ImageNet comparing ViTs and ConvNeXts. Extensive experiments show that minor changes in architecture, most notably replacing PatchStem with ConvStem, and training scheme have a significant impact on the achieved robustness. These changes not only increase robustness in the seen $\ell_\infty$-threat model, but even more so improve generalization to unseen $\ell_1/\ell_2$-attacks. Our modified ConvNeXt, ConvNeXt + ConvStem, yields the most robust $\ell_\infty$-models across different ranges of model parameters and FLOPs, while our ViT + ConvStem yields the best generalization to unseen threat models.

## 1 Introduction

Adversarial training [36] is the standard technique to obtain classifiers robust against adversarial perturbations. Many works have extensively analyzed several aspects of adversarial training, leading to further improvements. These include changes in the training scheme, i.e. using different loss functions [66, 43, 40], tuning hyperparameters like learning rate schedule or weight decay, increasing the strength of the attack at training time, adding label smoothing and stochastic weight averaging [19, 41]. Also, complementing the training set with pseudo-labelled [3] or synthetic [20] data, and via specific augmentations [45] yields more robust classifiers.

However, these works focus on small datasets like CIFAR-10 with low resolution images, and models using variants of ResNet [22] as architecture. When considering a more challenging dataset like ImageNet, the space of possible design choices regarding model, training and evaluation protocols becomes much richer, and little work exists exploring this domain. For example, it has been recently shown that architectures like vision transformers (ViTs) [16, 50], mixers [49, 54] and hybrids borrowing elements of different families of networks [17, 64] are competitive or outperform convolutional networks on ImageNet, while they hardly work on e.g. CIFAR-10. In turn, ConvNeXt [34] has been proposed as modern version of ResNet which closes the gap to transformer architectures.

In this work we study the influence of architecture and training schemes on the robustness of classifiers to seen and unseen attacks. We focus on two extreme ends of architectures used for ImageNet, ViT and ConvNeXt (isotropic vs non-isotropic, attention only vs convolution only, stem with large vs small patches), and study Isotropic ConvNeXt as an intermediate architecture. We focus on training ImageNet models robust with respect to the $\ell_\infty$-threat model (i.e. the perturbations have bounded $\ell_\infty$-norm), but additionally track the robustness to unseen $\ell_1$- and $\ell_2$-attacks in order to reveal architectural components which lead to better generalization of robustness. This, in fact, is a desirable

---

[*] Equal Contribution. Correspondence to: `naman-deep.singh@uni-tuebingen.de`

37th Conference on Neural Information Processing Systems (NeurIPS 2023).

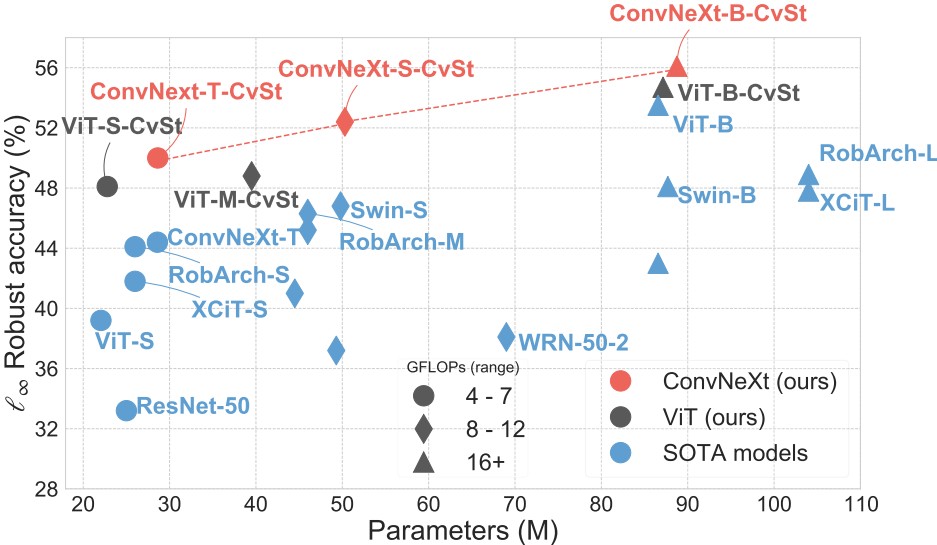

Figure 1: **AutoAttack $\ell_\infty$-Robust accuracy ($\epsilon = 4/255$) on ImageNet across different architectures and training schemes at resolution $224 \times 224$.** With our training scheme and replacing PatchStem with ConvStem in ConvNeXt and ViT, we outperform over a range of parameters/FLOPs existing models in terms of $\ell_\infty$-robust accuracy, e.g. our ConvNeXt-T-CvSt with 28.6M parameters achieves 50.2% robust accuracy, an improvement of 5.8% compared to the ConvNeXt-T of [14].

property since typically adversarially trained models are still vulnerable to attacks not seen at training time [28]. In particular, we show via extensive experiments that

- using a convolutional stem instead of a patch stem in most cases leads to consistent gains in $\ell_\infty$-robustness and quite a large boost in the unseen $\ell_1$- and $\ell_2$-threat models.
- leveraging SOTA pre-trained clean models as initialization allows us to train using heavy augmentation, in contrast to what has been suggested by prior work [2, 14].
- increasing image resolution at test time yields improvements in $\ell_\infty$-robust accuracy. This is surprising as for the $\ell_\infty$-threat model with fixed radius, higher resolution yields a stronger attack.

Combining our various improvements, we obtain state-of-the-art $\ell_\infty$-robust accuracy for perturbation size $\epsilon = 4/255$ at resolution $224 \times 224$ on ImageNet, see Fig. 1. We obtain 50.2% robust accuracy for small and 56.3% robust accuracy for large models, improving by 5.8% and 2.8% respectively upon prior works. Furthermore, we improve SOTA when fine-tuning these models to a larger radius, $\epsilon_\infty = 8/255$ and to other datasets. We make our code and robust models publicly available.[1]

## 2 Background and Related Work

In [36] adversarial training (AT) of a classifier $f_\theta : \mathcal{X} \to \mathcal{Y}$ is introduced as the optimization problem

$$\min_\theta \sum_{(x_i, y_i) \in \mathcal{D}} \max_{\delta : \|\delta\|_p \le \epsilon_p} \mathcal{L}(f_\theta(x_i + \delta), y_i) \tag{1}$$

where $\theta$ are the parameters of the classifier $f_\theta$. The goal is to make $f_\theta$ robust against $\ell_p$-bounded perturbations, that is the classifier should have the same prediction on the $\ell_p$-ball $B_p(x_i, \epsilon_p)$ of radius $\epsilon_p$ centered at $x_i$. In practice, the inner maximization is approximately solved with projected gradient descent (PGD), $\theta$ is optimized with SGD (or its variants) on the training set $\mathcal{D}$, and cross-entropy loss is used as objective function $\mathcal{L}$. AT has been shown to be an effective defense against adversarial perturbations, and many follow-up works have modified the original algorithm to further improve the robustness of the resulting models [66, 61, 20, 43, 40].

The most popular threat models are $\ell_\infty$- and $\ell_2$-bounded perturbations. Several other attacks have been explored, including $\ell_1$- or $\ell_0$-bounded, adversarial patches, or those defined by neural perceptual

---

[1] https://github.com/nmndeep/revisiting-at

metrics like LPIPS [30]. A well-known problem is that models trained to be robust with respect to one threat model need not be robust with respect to unseen ones [28]. Multiple-norm adversarial training [53] aims to overcome this but degrades the performance compared to adversarial training in each individual threat model.

**Training recipe.** [19, 41] analyzed in detail the influence of several fine-grained choices of training hyperparameters (batch size, weight decay, learning rate schedule, etc.) on the achieved robustness. [24, 5] suggest to use a standard classifier pre-trained either on a larger dataset or with a self-supervised task as initialization for adversarial training for improved robustness. However, all these works focus on CIFAR-10 or other datasets with low resolution images. In [38], the effect of several elements including clean initialization, data augmentation and length of training on adversarial training with ViTs is analyzed: however, their study is limited to CIFAR-10 and Imagenette [25], and we show below that some of their conclusions do not generalize to ImageNet and other architectures. For ImageNet, [2] argued that ViTs and ResNets attain similar robustness for several threat models if training scheme and architecture size for both are similar, while [14] proposed a recipe for training $\ell_\infty$-robust transformers, in particular XCiT [17]: basic augmentation, warm-up for the radius $\epsilon_p$, and large weight decay. In contrast, [44] use the standard training scheme for a large ViT-B for 300 epochs and obtain the most $\ell_\infty$-robust model on ImageNet. In concurrent work, [32] study $\ell_\infty$-training schemes for ConvNeXt and Swin-Transformer which are outperformed by our models, in particular regarding generalization to unseen attacks.

**Architectural design choices.** Searching for more robust networks [60, 26] studied the effect of architecture design choices in ResNets, e.g. width and depth, on adversarial robustness for CIFAR-10. Recently, [42] did a similar exploration on ImageNet. [48] compared the robustness to seen and unseen attacks of several modern architectures on CIFAR-10 and Imagenette, and conclude that ViTs lead to better robust generalization. Finally, [63] found that using smooth activation functions like GELU [23] improves the robustness of adversarially trained models on ImageNet, later confirmed for CIFAR-10 as well [19, 41].

**Modern architectures.** Since [16] showed that transformer-based architectures can be successfully used for vision tasks, a large number of works have introduced modifications of vision transformers [50, 33, 55] achieving SOTA classification accuracy on ImageNet, outperforming ResNets and variants. In turn, several works proposed variations of training scheme and architectures to make convolutional networks competitive with ViTs [47, 59], in particular leading to the ConvNeXt architecture of [34]. Moreover, hybrid models combine typical elements of ViTs and CNNs, trying to merge the strengths of the two families [17, 13, 64]. While these modern architectures perform similarly well on vision tasks, it is open how they differ regarding adversarial robustness. [18, 21] compared the robustness of normally trained ViTs and ResNets to $\ell_p$-norm bounded and patch attacks concluding that the ranking depends on the threat model.

# 3 Influence of Architecture on Adversarial Robustness to Seen and Unseen Attacks

In this section we analyze existing architectures and their components regarding adversarial robustness on ImageNet. In particular, we are interested in studying which elements are beneficial for adversarial robustness *(i)* in the threat model seen at training time and *(ii)* to unseen attacks. Since it is the most popular and well-studied setup, we focus on adversarial training w.r.t. $\ell_\infty$, and use $\ell_2$- and $\ell_1$-bounded attacks to measure the generalization of robustness to unseen threat models. While we focus on the influence of architecture on the achieved robustness in this section, we analyze the influence of pre-training and augmentation in Sec. 4. All results are summarized in Table 1 which provides the full ablation for this and the next section. We select the version T for ConvNeXt and S for Isotropic ConvNeXt and ViT, since these have comparable number of parameters and FLOPs.

## 3.1 Experimental Setup

**Training setup.** If not stated otherwise, we always use adversarial training [36] w.r.t. $\ell_\infty$ with perturbation bound $\epsilon = 4/255$ and 2 steps of APGD [8] for the inner optimization on ImageNet-1k using resolution 224x224 (we refer to App. C.2 for a comparison of APGD-AT vs PGD-AT). Note that we use the $\ell_2$- or $\ell_1$-threat model just for evaluation, we never do adversarial training for them. We use a cosine decaying learning rate preceded by linear warm-up. In this section, we use a basic

training setting, that is only random crop as basic augmentation and 50 epochs of adversarial training from random initialization. More details are available in App. A.

**Evaluation setup.** For all experiments in the paper, unless specified otherwise, we evaluate the robustness using AutoAttack [8, 9], which combines APGD for cross-entropy and targeted DLR loss, FAB-attack [7] and the black-box Square Attack [1], on the 5000 images of the ImageNet validation set selected by RobustBench [6] at resolution 224x224. The evaluations are done at the following radii: $\epsilon_\infty = 4/255$, $\epsilon_2 = 2$, and $\epsilon_1 = 75$. For some marked experiments which involve a large number of evaluations we use $\text{APGD}_{\text{T-DLR}}$ (40 iterations, 3 restarts) with the targeted DLR loss [8] which is roughly off by 1-2% compared to full AutoAttack.

## 3.2 Architecture Families and Convolutional Stem

**Choice of architectures.** While some of the novel components of ConvNeXts are inspired by the Swin Transformer [33], the ConvNeXt is a modified version of a ResNet, using progressive downsampling and convolutional layers. On the other end of the spectrum of network families we find ViTs, as a non-convolutional architecture using attention and with an isotropic design after the PatchStem, see Fig. 2. Additionally, [34] introduce Isotropic ConvNeXt, which creates tokens similarly to ViTs, which are then processed by a sequence of identical residual blocks (with no downsampling, hence the name isotropic) as in ViTs, until a final linear classifier. The blocks of a ViT rely on multi-head self-attention layers [56], while Isotropic ConvNeXt replaces them with depthwise convolutions. It can be thus considered an intermediate step between ConvNeXt and ViT. By considering ViT and ConvNeXt, we cover two extreme cases of modern architectures. While it would have been interesting to include other intermediate architectures like XCiT or Swin-Transformer, our computational budget did not allow this on top of the already extensive experiments of the present paper.

**Convolutional stem.** [62] propose to replace the patch embedder in ViTs with a convolutional block (ConvStem) consisting of several convolutional layers alternating with normalization and activation layers. While the PatchStem in ViTs divides the input in disjoint patches and creates tokens from them, the ConvStem progressively downsamples the input image until it has the right resolution to be fed to the transformer blocks. [62] show that this results in a more stable standard training which is robust to the choice of optimizer and hyperparameters. As adversarial training is more difficult than standard training, we check if adding a ConvStem has a similar positive impact there as well. As both ViTs and Isotropic ConvNeXts divide the input image into $16 \times 16$ patches, we use for both the convolutional stem of [62], which has four convolutional layers with stride 2, each followed by layer normalization and GELU activation. Since ConvNeXt uses a PatchStem with $4 \times 4$-patches, the ConvStem of ViTs would not work. Therefore we design a specific convolutional stem with only two convolutional layers with kernel size 3 and stride 2, see Fig. 2. Adding a ConvStem has only minimal impact on the number of parameters and FLOPs of the models (see Table 2), especially for ConvNeXt.

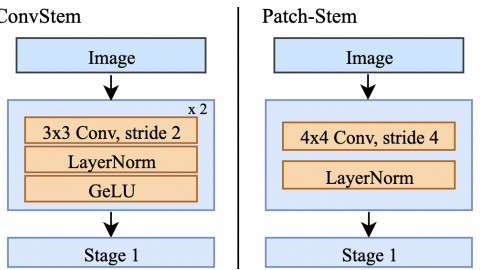

Figure 2: **Convolutional stem for ConvNeXt.** We use two convolution layers, each followed by LayerNorm and GELU to replace the 4x4 patch of the standard ConvNeXt model (PatchStem).

More details on the convolutional stem design, and ablation studies on its effect on unseen threat models can be found in App. B. The effect of ConvStem on model size is discussed in App. C.4.

## 3.3 Effect of Architecture and ConvStem vs Patchstem on Adversarial Robustness

In Table 1 we first check for the basic setting described above (random init., basic augment.) the effect of using a convolutional stem (+ CvSt) vs PatchStem for the three architectures: for both isotropic architectures it notably improves the robustness as well as clean accuracy for adversarially trained models, whereas the improvements for the ConvNeXt are marginal. However, all three architectures show huge improvements in the generalization of robustness to unseen $\ell_1$- and $\ell_2$-attacks when using ConvStem instead of PatchStem. Interestingly, the $\ell_1$- and $\ell_2$-robustness of the ConvStem models is quite similar despite larger differences in $\ell_\infty$-robustness. Our results in this basic training setup show

Table 1: **Influence of ConvStem, strong (clean) pre-training and heavy data augmentation.** We show an ablation of using strong (clean) pre-training, heavy data augmentation and longer training for all models with and without ConvStem. Across all architectures using ConvStem, stronger pre-training, heavy augmentation and longer training improve adversarial robustness. We boldface the best performance for every metric within each architecture family.

| Architecture | Training Scheme | Adversarial Training w.r.t. $\ell_\infty$ | | | |
| --- | --- | --- | --- | --- | --- |
| | | clean | $\ell_\infty$ | $\ell_2$ | $\ell_1$ |
| ConvNeXt-T | random init., basic augment. | 64.0 | 37.1 | 28.9 | 8.8 |
| | + strong clean pre-training | 69.4 +5.4 | 41.6 +4.5 | 42.4 +13.5 | 18.9 +10.1 |
| | + heavy augmentations | 71.0 +1.6 | 46.5 +4.9 | 38.1 -4.3 | 14.9 -4.0 |
| | 50 → 300 epochs | 72.4 +1.4 | 48.6 +2.1 | 38.0 -0.1 | 14.9 +0.0 |
| ConvNeXt-T + CvSt | random init., basic augment. | 64.2 | 37.6 | 40.2 | 18.6 |
| | + strong clean pre-training | 69.1 +4.9 | 42.2 +4.6 | 42.4 +2.2 | 19.7 +1.1 |
| | + heavy augmentations | 69.1 +0.0 | 47.5 +5.3 | 47.8 +5.4 | **24.6** +4.9 |
| | 50 → 300 epochs | **72.7** +3.6 | **49.5** +2.0 | **48.4** +0.6 | 24.5 -0.1 |
| Isotropic ConvNeXt-S | random init., basic augment. | 60.5 | 31.7 | 20.8 | 4.0 |
| | + strong clean pre-training | 68.0 +7.5 | 38.9 +7.2 | 38.3 +17.5 | 17.0 +13.0 |
| | + heavy augmentations | 64.5 -3.5 | 38.5 -0.4 | 36.5 -1.8 | 17.7 +0.7 |
| | 50 → 300 epochs | 69.0 +4.5 | 44.2 +5.7 | 36.6 +0.1 | 14.9 -2.8 |
| Iso-ConvNeXt-S + CvSt | random init., basic augment. | 62.1 | 33.5 | 40.0 | 23.8 |
| | + strong clean pre-training | 69.5 +7.4 | 41.8 +8.3 | 46.2 +6.2 | 26.9 +3.1 |
| | + heavy augmentations | 69.5 +0.0 | 42.1 +0.3 | 47.2 +1.0 | **28.9** +2.0 |
| | 50 → 300 epochs | **70.2** +0.7 | **45.9** +3.8 | **49.2** +2.0 | 27.9 -1.0 |
| ViT-S | random init., basic augment. | 61.5 | 31.8 | 35.5 | 15.1 |
| | + strong clean pre-training | 66.8 +5.3 | 39.1 +7.3 | 34.5 -1.0 | 12.6 -2.5 |
| | + heavy augmentations | 65.2 -1.6 | 39.2 +0.1 | 37.3 +2.8 | 16.2 +3.6 |
| | 50 → 300 epochs | 69.2 +4.0 | 44.0 +4.8 | 37.5 +0.2 | 15.1 -1.1 |
| ViT-S + CvSt | random init., basic augment. | 62.8 | 34.4 | 39.2 | 20.4 |
| | + strong clean pre-training | 71.2 +8.4 | 44.3 +9.9 | 47.1 +7.9 | 23.2 +2.8 |
| | + heavy augmentations | 69.9 -1.3 | 44.0 -0.3 | 47.1 +0.0 | 25.9 +2.7 |
| | 50 → 300 epochs | **72.5** +2.6 | **48.1** +4.1 | **50.4** +3.3 | **26.7** +0.8 |

that the ConvStem is a simple and effective architecture modification which has significant impact on robustness with respect to seen and unseen threat models. We also note that this finding generalizes for the modifications of the training scheme proposed in the next section. It is particularly remarkable that even for the ConvNeXt architecture, this little change from PatchStem to ConvStem has such a huge effect on the robustness with respect to the unseen $\ell_1$-and $\ell_2$-threat models.

## 4 Effect of Strong Pre-Training and Heavy Data Augmentation on Robustness

As the next step after the architecture modification of the last section, we now identify in an ablation study in Table 1 strong pre-training and heavy augmentations as key elements of the training scheme which boost adversarial robustness on ImageNet. The effect of other fine-grained training hyperparameters like weight decay and label smoothing is discussed in App. D.3, where we show that varying them around our default values does not have any impact on robustness.

### 4.1 Strong Pre-Training

It is well-known that pre-training on large datasets improves the performance on downstream tasks, e.g. models pre-trained on ImageNet-21k or JFT achieve significantly higher accuracy on ImageNet-1k than those trained on ImageNet-1k from random initialization [58]. Similarly, it has been observed that pre-training on larger datasets like ImageNet-21k or ImageNet-1k helps to achieve better adversarial robustness on CIFAR-10 [24, 38]. However, our goal is not to use a larger training set than ImageNet-1k but to better utilize the given dataset ImageNet-1k. As adversarial training on ImageNet is a non-trivial optimization problem, where runs can fail completely, see [14], we propose to initialize adversarial training from a fully trained (on ImageNet-1k) standard model. Although the clean model

is clearly not adversarially robust, starting from a point with low clean loss considerably helps in the first stages of adversarial training. Short fine-tuning of clean models to become adversarially robust has been discussed in [27, 10] but without achieving robustness comparable to full adversarial training.

In the following we use as initialization the standard models available in the `timm` library or from the original papers, refer App. D.7 for further details. For the models with convolutional stem no such models are available. For these we initialize the ConvStem randomly and the remaining layers with the weights of the corresponding non-ConvStem models, then do standard training for 100 epochs to reach good clean accuracy (see App. B.3 for details and performance of these classifiers). Interestingly, changing from random initialization to initialization with strongly pre-trained models significantly improves clean as well as $\ell_\infty$-robust accuracy across architectures, see Table 1. The effects on the isotropic architectures, which are known to be harder to train, are even higher than for the ConvNeXt. At this stage the ViT-S + ConvStem already has 2.5% better $\ell_\infty$-robustness (44.3% vs 41.8%) than the XCiT-S model of [14]. Finally, using an even better model, i.e. pre-trained on ImageNet-21k, as initialization did not provide additional benefit, see App. D.1.

## 4.2 Heavy Data Augmentation

The second aspect we analyze is how to take advantage of heavy augmentation techniques like RandAugment [12], MixUp [67] and CutMix [65], which are crucial for the performance of standard models, in adversarial training. In particular, [2] note that for ViTs using heavy data augmentation with adversarial training leads to model collapse, and thus design a schedule to progressively increase the augmentation intensity. Similarly, [14] report better performance for robust ViTs when only weak augmentations (random crop, horizontal flip, color jitter) are used, and that ConvNeXt training even diverges if heavy augmentations are not excluded. Finally, [44] could train with heavy augmentations but using a much larger ViT-B model and a longer training of 300 epochs (which allows them to have a less steep learning rate warm-up phase). Instead we show that initializing adversarial training with a well-trained standard classifier makes it possible to use heavy augmentation techniques from the beginning, and benefit from them, even for small models and shorter training schedules. In contrast, in our experiments, training a ConvNeXt with heavy augmentations from random initialization failed.

As heavy augmentations we use RandAugment, CutMix, MixUp, Random Erasing [68] (with hyperparameters similar to [34], see App. A). We also use Exponential Moving Average (EMA) with a decay of 0.9999 and label smoothing with parameter 0.1 for consistency with [34]. However, we observe little to no effect of EMA on robustness.

In Table 1 one can see that adding heavy data augmentation on top of strong pre-training significantly boosts the performance of the ConvNeXts in clean and $\ell_\infty$- robust accuracy. Interestingly, only the ConvNeXt + ConvStem has further strong improvements in the unseen $\ell_2$- and $\ell_1$-threat models. For the isotropic architectures, heavy augmentations have marginal effect on $\ell_\infty$-robustness, although they improve robustness to unseen attacks. Isotropic architectures require longer training to benefit from heavy augmentations. We highlight that at this point our ConvNeXt-T and ConvNeXt-T + ConvStem already have higher (+2.1% and +3.1% respectively) robustness w.r.t. $\ell_\infty$ than the ConvNeXt-T from [14], and similarly our ViT-S + ConvStem and Isotropic ConvNeXt + ConvStem outperform XCiT-S [14] (detailed comparison to SOTA in Table 2). A further comparison of heavy augmentation and the 3-Aug scheme of [14] can be found in in App. D.2.

**Longer training.** Adding heavy data augmentation increases the complexity of the learning problem, which requires longer training, in particular for isotropic architectures. Thus we extend the training time from 50 to 300 epochs. As expected we see the strongest improvements for the isotropic architectures but even for ConvNeXt one observes consistent gains: this is in contrast to the claim of [38] that longer training does not help to improve robustness. The ConvNeXt-T + ConvStem, with 49.5% robust accuracy, outperforms all classifiers reported in [14] and [42], including those with up to 4 times more parameters.

## 5 Comparison with SOTA Models

In Table 2 we compare our classifiers with models from prior works trained for adversarial robustness w.r.t. $\ell_\infty$ at $\epsilon_\infty = 4/255$ on ImageNet. For each model we report number of parameters and FLOPs,

Table 2: **Comparison to SOTA $\ell_\infty$-robust models on ImageNet.** For each model we report the number of parameters, FLOPs, number of epochs of adversarial training (AT), the number of PGD steps in AT, clean and $\ell_\infty, \ell_2, \ell_1$-robust accuracy with $\epsilon_\infty = 4/255$, $\epsilon_2 = 2$, $\epsilon_1 = 75$ (AutoAttack). Models of similar size are sorted in increasing $\ell_\infty$-robustness. Our ConvNeXt + ConvStem improves by 5.8% for small, 7.2% for medium, and 2.8% for large models over SOTA.

| | Architecture | Params (M) | FLOPs (G) | Source | Ep. | Adv. Steps | Adversarial Tr. wrt $\ell_\infty$ | | | |
|---|---|---|---|---|---|---|---|---|---|---|
| | | | | | | | clean | $\ell_\infty$ | $\ell_2$ | $\ell_1$ |
| Small | ResNet-50 | 25.0 | 4.1 | [46] | 100 | 3 | 65.88 | 33.18 | 18.88 | 3.82 |
| | ResNet-50 | 25.0 | 4.1 | [2] | 100 | 1 | 67.44 | 35.54 | 18.16 | 3.90 |
| | ViT-S | 22.1 | 4.6 | [2] | 100 | 1 | 66.62 | 36.56 | 41.40 | 21.82 |
| | ViT-S | 22.1 | 4.6 | [14] | 110 | 1 | 66.78 | 37.88 | – | – |
| | ViT-S | 22.1 | 4.6 | [37] | 90 | 3 | 65.9 | 39.24 | 32.18 | 10.54 |
| | XCiT-S12 | 26.0 | 4.8 | [14] | 110 | 1 | 72.34 | 41.78 | 46.20 | 22.72 |
| | ViT-S | 22.1 | 4.6 | ours | 300 | 2 | 69.22 | 44.04 | 37.52 | 15.12 |
| | RobArch-S | 26.1 | 6.3 | [42] | 110 | 3 | 70.58 | 44.12 | 39.88 | 15.46 |
| | Isotropic-CN-S | 22.3 | 4.3 | ours | 300 | 2 | 69.04 | 44.22 | 36.64 | 14.88 |
| | ConvNeXt-T | 28.6 | 4.5 | [14] | 110 | 1 | 71.60 | 44.40 | 45.32 | 21.76 |
| | Isotropic-CN-S + ConvStem | 23.0 | 4.7 | ours | 300 | 2 | 70.02 | 45.90 | 49.24 | **27.84** |
| | ViT-S + ConvStem | 22.8 | 5.0 | ours | 300 | 2 | 72.56 | 48.08 | **50.40** | 26.68 |
| | ConvNeXt-T | 28.6 | 4.5 | ours | 300 | 2 | 72.40 | 48.60 | 38.02 | 14.88 |
| | ConvNeXt-T + ConvStem | 28.6 | 4.6 | ours | 300 | 2 | **72.72** | 49.46 | 48.42 | 24.52 |
| | ConvNeXt-T + ConvStem | 28.6 | 4.6 | ours | 300 | 3 | 72.70 | **50.16** | 49.00 | 24.16 |
| Medium | Wide-ResNet-50-2 | 68.9 | 11.4 | [46] | 100 | 3 | 68.82 | 38.12 | 22.08 | 4.48 |
| | ResNet-101 | 44.5 | 7.9 | [37] | 90 | 3 | 69.52 | 41.02 | 25.62 | 6.56 |
| | XCiT-M12 | 46.0 | 8.5 | [14] | 110 | 1 | 74.04 | 45.24 | 48.18 | 22.72 |
| | ViT-M | 38.8 | 8.0 | ours | 50 | 2 | 71.72 | 47.24 | 49.02 | **29.20** |
| | ViT-M + ConvStem | 39.5 | 8.4 | ours | 50 | 2 | 72.40 | 48.80 | 50.56 | 28.12 |
| | ConvNeXt-S | 50.1 | 8.7 | ours | 50 | 2 | **74.10** | 52.32 | 43.84 | 19.52 |
| | ConvNeXt-S + ConvStem | 50.3 | 8.8 | ours | 50 | 2 | **74.10** | 52.42 | **50.88** | 25.64 |
| Large | ViT-B | 86.6 | 17.6 | [37] | 90 | 3 | 70.42 | 43.02 | 47.26 | 27.08 |
| | XCiT-L12 | 104.0 | 19.0 | [14] | 110 | 1 | 73.76 | 47.60 | 49.38 | 23.74 |
| | Swin-B | 87.7 | 15.5 | [37] | 90 | 3 | 74.76 | 48.10 | 44.42 | 18.04 |
| | RobArch-L | 104.0 | 25.7 | [42] | 100 | 3 | 73.46 | 48.92 | 39.48 | 14.74 |
| | ViT-B | 86.6 | 17.6 | ours | 50 | 2 | 73.32 | 50.02 | 52.14 | **33.12** |
| | ViT-B + ConvStem | 87.1 | 17.9 | ours | 50 | 2 | 74.38 | 52.58 | 54.38 | 31.20 |
| | ViT-B | 86.6 | 17.6 | [44] | 300 | 2 | **76.62** | 53.50 | – | – |
| | ConvNeXt-B | 88.6 | 15.4 | ours | 50 | 2 | 75.62 | 54.34 | 48.52 | 23.70 |
| | ConvNeXt-B + ConvStem | 88.8 | 16.0 | ours | 50 | 2 | 75.32 | 54.38 | 50.06 | 24.76 |
| | ViT-B + ConvStem | 87.1 | 17.9 | ours | 250 | 2 | 76.30 | 54.66 | **56.30** | 32.06 |
| | ConvNeXt-B | 88.6 | 15.4 | [32]* | 300 | 3 | 76.02 | 55.82 | 44.68 | 21.23 |
| | ConvNeXt-B + ConvStem | 88.8 | 16.0 | ours | 250 | 2 | 75.90 | 56.14 | 49.12 | 23.34 |
| | Swin-B | 87.7 | 15.5 | [32]* | 300 | 3 | 76.16 | 56.16 | 47.86 | 23.91 |
| | ConvNeXt-B + ConvStem | 88.8 | 16.0 | ours | 250 | 3 | 75.18 | **56.28** | 49.40 | 23.60 |

* concurrent work

the number of training epochs, the number of steps in the inner maximization, and clean and robust accuracy (AutoAttack on RobustBench validation set) in the three threat models. We group the models according to size for fair comparison, distinguishing small ($< 30$M parameters), medium ($\approx 50$M params) and large models ($\geq 80$M params).

**1-Step vs 2-Step Adversarial Training.** Current works use one to three steps of PGD to solve the inner problem, see column "Adv. Steps" in Table 2. It is an important question if our gains, e.g. compared to works like [14] who use a single step, are just a result of the stronger attack at training time. We test this by training a ConvNeXt-T + ConvStem model with strong pre-training and heavy augmentation with 1-step APGD adversarial training for 50 epochs. We get a $\ell_\infty$-robust accuracy

of 46.4% (clean acc. is 71.0%) compared to 47.5% with 2-step APGD (see Table 1) which is just a difference of 1.1%. Thus our gains are only marginally influenced by stronger adversarial training. Moreover, our ConvNeXt + ConvStem models with 50 epochs of adversarial training outperform all existing SOTA models even though the computational cost of training them is lower (50 epochs of 2-step AT corresponds roughly to 75 epochs of 1-step AT and 37.5 epochs of 3-step AT). For our best small and large model we also check the other direction and train 3-step AT models, which improve by 0.7% for ConvNeXt-T + ConvStem and 0.14% for ConvNeXt-B + ConvStem in $\ell_\infty$-robustness. This suggests that 2-step AT is a good compromise between achieved robustness and training time. In App. C.1 we additionally show how our training scheme yields more robust models than that of [14], even with less than half the effective training cost.

**Scaling to larger models.** We apply our training scheme introduced in the previous sections to architectures with significantly more parameters and FLOPs. We focus on ConvNeXt-S/B and ViT-M/B: in particular, for ConvNeXt-B we adapt the convolutional stem used for ConvNeXt-T by adding an extra convolutional layer, while for the ViTs we just adjust the number of output channels to match the feature dimension of the transformer blocks (more details in App. B). We train the medium models for 50 epochs and the large models for 50 and 250 epochs (with ConvStem only).

**Comparison with SOTA methods.** Among small models, all three architectures equipped with ConvStem outperform in terms of $\ell_\infty$-robustness all competitors of similar size with up to 5.8% improvement upon the ConvNeXt-T of [14]. Moreover, our ConvNeXt-T + ConvStem is with 50.2% robust accuracy 1.2% better than the RobArch-L [42] (the previously second most robust model) which has 3.6× more parameters and FLOPs. The ConvStem positively impacts generalization of robustness to unseen attacks of all three architectures, with improvements between 2.8% and 4.2% for $\ell_2$ and 1.4% and 5.2% for $\ell_1$ (with respect to the XCiT-S12). For medium models the picture is the same: our ViT-M+ConvStem and the ConvNeXt-S+ConvStem outperform both in seen and unseen threat models their non-ConvStem counterparts and the best existing model (XCiT-M12 of [14]).

For large models we compare to the previously most robust model, the ViT-B of [44], with 53.5% $\ell_\infty$-robust accuracy (the model is not available so we cannot evaluate $\ell_1$- and $\ell_2$-robustness). It is remarkable that our ConvNeXt-B+ConvStem with only 50 epochs of training outperforms this ViT-B trained for 300 epochs already by 0.9% and with 250 epochs training even by 2.5%. An interesting observation is that our ViT-B+ConvStem with 250 epochs training is with 54.7% just 1.3% worse than our ConvNeXt-B+ConvStem but has significantly better generalization to unseen attacks: 56.3% vs. 49.1% for $\ell_2$ and 32.1% vs 23.3% for $\ell_1$. Regarding the concurrent work of [32], we note that our ConvNeXt-B+ConvStem trained for 250 epochs with 2-step AT is already better than their ConvNeXt-B 3-step AT model trained for 300 epochs ($\approx$ 40% longer training time), while having significantly better generalization to the unseen $\ell_1$- and $\ell_2$-attacks. Our 3-step AT ConvNeXt-B+ConvStem outperforms their best model, a Swin-B, even though it is trained for 20% less epochs, and has better generalization to $\ell_2$. Finally, we note that our ViT-B+ConvStem significantly outperforms their Swin-B in terms of robustness to unseen attacks.

**Additional experiments.** InternImage [57], a recent non-isotropic model, already uses a ConvStem. Replacing it with a PatchStem preserves the robustness in $\ell_\infty$ (seen threat model) but decreases the robustness to unseen $\ell_1$- and $\ell_2$-attacks (App. C.3), further validating the effectiveness of ConvStem.

Table 3: **Optimal test time image resolution.** For our small and large models trained at 224×224 with 300 or 250 epochs AT, we provide the optimal test time resolution and the resulting improvements (green) in clean and $\ell_\infty$-robust accuracy ($\epsilon$ =4/255) compared to the training resolution.

| Architecture | Res. | clean | | $\ell_\infty$ | |
|---|---|---|---|---|---|
| ConvNeXt-T | 256 | 72.4 | +0.0 | 49.4 | +0.8 |
| ConvNeXt-T + CvSt | 288 | 74.8 | +1.9 | 50.8 | +1.3 |
| Iso-CN-S | 288 | 71.2 | +2.2 | 46.4 | +2.2 |
| Iso-CN-S + CvSt | 288 | 72.3 | +2.1 | 48.1 | +2.2 |
| ViT-S | 256 | 69.9 | +0.7 | 44.2 | +0.2 |
| ViT-S + CvSt | 288 | 73.7 | +1.2 | 49.3 | +1.2 |
| ViT-B + CvSt | 256 | 76.6 | +0.3 | 55.8 | +1.1 |
| ConvNeXt-B + CvSt | 256 | 76.9 | +1.0 | 57.3 | +1.2 |

Table 4: **ImageNet: $\ell_\infty$-robust accuracy at $\epsilon_\infty$ = 8/255.** While models from the literature are trained from scratch for $\epsilon_\infty$ = 8/255, we fine-tune our 4/255-robust models for only 25 epochs to the larger radius.

| Architecture | Source | clean | $\ell_\infty$ |
|---|---|---|---|
| ResNet-50 | [46] | 54.9 | 19.7 |
| XCiT-S | [14] | 63.5 | 25.0 |
| ViT-S + CvSt | ours | 67.5 | 28.2 |
| XCiT-L | [14] | 69.2 | 28.7 |
| CN-T + CvSt | ours | 68.6 | 29.6 |
| CN-B + CvSt | ours | **71.7** | **33.2** |

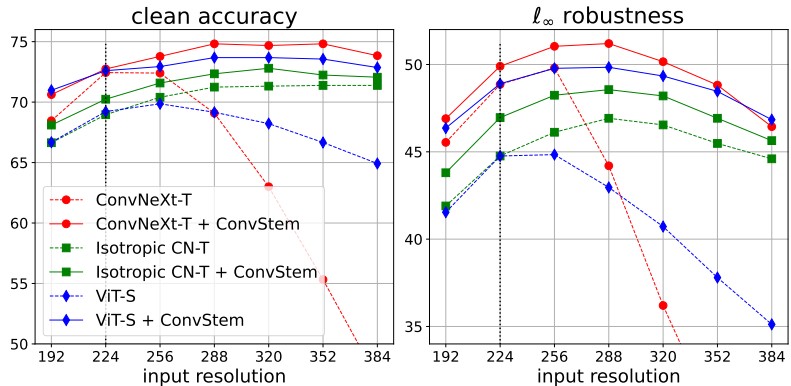

Figure 3: **(Robust) accuracy improves with increasing test-time image resolution.** Evaluation of clean (left) and $l_\infty$-robust accuracy (right, evaluation using APGD$_{\text{T-DLR}}$) at $\epsilon_\infty = 4/255$ for the top models from Table 1 for varying test input resolution.

## 6 Influence of Resolution and Fine-Tuning

In the following we analyze how robustness behaves under variation of test time resolution, and robust fine-tuning to either a larger radius on ImageNet or other datasets.

**Improving robustness via increasing test-time image resolution.** Unlike small datasets like CIFAR-10, for ImageNet the input images are resized before inference. Using higher resolution at test time can lead to improved clean accuracy for standard ImageNet models [52, 17]. On the other hand increasing dimension for fixed radius makes the $\ell_\infty$-threat model more powerful. To study which effect prevails, we test clean and $\ell_\infty$-robust accuracy for each of our 2-step AT models from 'small' (300 epochs) and 'large' (250 epochs) in Table 2 when varying test time input resolution between 192 and 384. Note that all models have been trained on 224x224 images only, and no fine-tuning to different resolutions is done. Also, for ViTs we have to adapt the models via interpolation of the positional embedding as suggested by [4]. Fig. 3 shows that all classifiers obtain their highest robustness for image resolutions of 256 or 288, higher than the training one. For faster evaluation we use for this plot APGD$_{\text{T-DLR}}$ with 40 iterations and 3 target classes. Then, we select for each model the best input resolution, and evaluate $\ell_\infty$-robustness with AutoAttack in Table 3. Increasing input resolution significantly improves clean and $\ell_\infty$-robust accuracy. Our ConvNeXt-B + ConvStem attains SOTA 76.9% and 57.3% clean and $\ell_\infty$-robust accuracy, respectively, at resolution 256. Additional results on the role of the input resolution can be found in App. D.5, including a discussion of what it means for threat models other than $\ell_\infty$.

**Varied perturbation strength.** In Table 5, we test the effect of varying the test time resolution on the robustness in $\ell_\infty$ for radii other than $4/255$ used for training (we use our ConvNeXt-B + ConvStem, with robustness evaluation via AutoAttack on the RobustBench ImageNet set). For all radii $\epsilon_\infty$, the best resolution is either 256 or 288, that is larger than the training resolution of 224, which supports

Table 5: **Increased resolution across perturbation strengths.** For ConvNeXt-B+ConvStem, we see that both best clean and $\ell_\infty$ robust accuracies are attained a a higher resolution than the one trained for (224) across all values of $\epsilon_\infty$. The difference from base value at the resolution of 224 is shown in color and the best result for each perturbation strength (row) is **highlighted**.

| $\epsilon_\infty$ | Input resolution | | | | | | | | | |
|---|---|---|---|---|---|---|---|---|---|---|
| | 192 | | 224* | 256 | | 288 | | 320 | |
| clean | 74.1 | -1.8 | 75.9 | 76.9 | +1.0 | **77.7** | +1.8 | 77.2 | +1.3 |
| 2/255 | 64.6 | -2.3 | 66.9 | 67.9 | +1.0 | **68.6** | +1.7 | 68.4 | +1.5 |
| 4/255 | 53.0 | -3.2 | 56.1 | **57.3** | +1.2 | 57.2 | +1.1 | 56.6 | +0.5 |
| 6/255 | 41.0 | -2.8 | 43.8 | 44.4 | +0.6 | **44.5** | +0.7 | 43.0 | -0.8 |
| 8/255 | 29.5 | -0.9 | 30.4 | **31.0** | +0.6 | 29.8 | -0.6 | 27.9 | -2.5 |

Table 6: $\ell_\infty$-**Robust accuracy at radius** $\epsilon_\infty = 8/255$. We examine fine-tuning our ConvNeXt + ConvStem models robust to $\epsilon_\infty = 8/255$ from Table 4 to other datasets. We compare to the XCiT models of [14] when available.

| Architecture | Flowers-102 | | CIFAR-100 | | CIFAR-10 | |
|---|---|---|---|---|---|---|
| | clean | $\ell_\infty$ | clean | $\ell_\infty$ | clean | $\ell_\infty$ |
| XCiT-S | **82.86** | 47.91 | 67.34 | 32.19 | 90.06 | **56.14** |
| CN-T + CvSt | 80.42 | **50.89** | **72.32** | **32.30** | **92.72** | 55.97 |
| XCiT-L | – | – | 70.76 | **35.08** | 91.73 | 57.58 |
| CN-B + CvSt | 83.98 | 54.77 | **74.83** | 34.24 | **93.55** | **58.17** |

the results above. Unlike for the smaller radii, at $\epsilon_\infty = 8/255$ the robust accuracy at both resolutions 288 and 320 is worse than at 224. This suggests that there are diminishing returns for higher radii. The attack strength increases more quickly with the perturbation budget at higher resolution than the improvements in clean accuracy. While for the $\ell_\infty$-threat model the comparison across different image resolutions can be done directly, the comparison for the $\ell_2$-threat model is more subtle and we discuss potential ways to do this in App. D.5. If one keeps the perturbation budget per pixel constant, that means increasing $\epsilon_2$ linearly with the image resolution, then the gains in $\ell_2$-robustness are marginal.

**Fine-tuning to larger radius.** While $\epsilon = 4/255$ is the standard radius for the $\ell_\infty$-threat model on ImageNet, even larger perturbations might still be imperceptible. [14] train XCiTs with $\epsilon = 8/255$ from scratch for 110 epochs. However, given the success of using a warm start for adversarial training (see Sec. 4.1), we propose to fine-tune the model fully trained for robustness at $\epsilon = 4/255$ with adversarial training with the larger radius. Such classifiers already have non-trivial robustness at $\epsilon = 8/255$, and we expect they require only relatively small changes to adapt to the larger radius. We use our models from Table 2 to fine-tune for 25 epochs (details in App. A.5), and report the results in Table 4: our ConvNeXt-B + ConvStem improves significantly over the previous best result both in terms of clean and robust accuracy, even though we just use relatively short fine-tuning.

**Fine-tuning to other datasets.** We fine-tune our models trained to be robust at radius $8/255$ from Table 4 to other datasets and compare to the results of [14] in Table 6. For Flowers-102 (resolution $224 \times 224$) the improvement over XCiT in robust accuracy is similar to the difference of the models in Table 4 and shows that the benefits of more robust models transfer to smaller datasets. For CIFAR (resolution $32 \times 32$) our models tend to achieve similar robustness as the XCiT models but at a much better clean accuracy. We expect that here even stronger robustness gains are possible by further optimizing the fine-tuning. For details on the fine-tuning setup see App. D.6.

## 7 Conclusion

We have examined two extreme representatives of today's modern architectures for image classification, ViT and ConvNeXt, regarding adversarial robustness. We have shown that ConvNeXt + ConvStem yields SOTA $\ell_\infty$-robust accuracy as well as good generalization to unseen threat models such as $\ell_1$ and $\ell_2$ across a range of parameters and FLOPs. Given that for smaller model sizes the ConvNeXt + ConvStem is much easier to train than the ViT, it has the potential to become similarly popular for studying adversarial robustness on ImageNet like (Wide)-ResNets for CIFAR-10. On the other hand ViT + ConvStem excels in generalization to unseen threat models. Thus there is no clear winner between ViT and ConvNeXt. Given that ImageNet backbones are used in several other tasks, we think that our robust ImageNet models could boost robustness also in these domains. One such example is our recent work in robust semantic segmentation [11]. Another interesting open question is why the ConvStem leads to such a boost in generalization to $\ell_1$- and $\ell_2$-threat models.

**Limitations.** Given the extensive set of experiments already presented in this work and our limited computational resources, we were unable to include other modern architectures. Hence, even though we cover quite diverse architectures, we do not cover all modern architectures for ImageNet.

**Broader Impact.** As this work studies the robustness of deep neural networks to adversaries, we do not see any potential negative impact of our work. Instead, our work could help make modern deep models robust and safer to deploy.

## Acknowledgements

The authors acknowledge support from the DFG Cluster of Excellence "Machine Learning – New Perspectives for Science", EXC 2064/1, project number 390727645 and Open Philanthropy. The authors thank the International Max Planck Research School for Intelligent Systems (IMPRS-IS) for supporting Naman D Singh.

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

## Contents of the Appendix

## A   Experimental Details

In the following we provide the details relative to the setup used for the various experiments. First, we describe the general setup (App. A.1), then detail the variations for different settings, if they differ from the main regime.

### A.1   General experimental setup

In all the experiments on ImageNet [15], unless stated otherwise, we use training parameters similar to [44]. All models are trained with AdamW [35] as optimizer with a cosine decay scheduler for the learning rate (LR), weight decay of 5e-2 and momenta terms $\beta_1$ and $\beta_2$ set to 0.9 and 0.95 respectively. We employ a peak LR of 1e-3 (chosen to be in the same range as the values of the original ConvNeXt [34] and DeiT [51] works), slightly higher than [44]. When training for 50 or 100 epochs the peak LR is attained at epoch 10, when using 250 or 300 epochs at epoch 20. We also use label smoothing coefficient of 0.1 and exponential moving average (EMA) with decay rate 0.999. For the inner maximization in adversarial training we use 2-step APGD [8] with an $\ell_\infty$ radius of $4/255$ and cross entropy as objective function.

**Computational expense.** All the training runs were conducted on a multi-GPU (8) setup using NVIDIA A100s with 40GB memory. Around 20,000 hours of single-GPU run-time was expended during this work.

**ConvStem models clean training.** All the models with ConvStem (barring the one in 'random init., basic augment.' row in Table 1) were normally (non-adversarially) trained for 100 epochs with the non-ConvStem part of the model initialized with ImageNet-1k pre-trained weights from `timm`. Heavy augmentations (App. A.2) were used for this pre-training. For 'medium' and 'large' category models in Table 2, we only did standard training for 50 epochs to save computational load given the network size, still achieving similar clean performance as their non-ConvStem counterparts initialized with weights from `timm` (see Table 10). The checkpoint with highest accuracy was then used as the starting point for adversarial training. The specific `timm` configuration for pre-trained weights for every model is available in Table 18.

**Robustness evaluation.** Unless stated otherwise, all robustness evaluations are done using complete standard AutoAttack [8, 9] (includes 4 different attacks) on the complete ImageNet validation set (5k points) from RobustBench [6]. As is the common practice, we do the best (most robust) checkpoint selection in a post-hoc fashion using 1-step APGD on a separate validation set of 4k ImageNet-1k images. Note that we use a validation set disjoint from the RobustBench ImageNet test set.

**Calculation of model size.** We compute the number of FLOPs and parameters using the `fvcore`[2] library. In further sections, we again differentiate models as 'small' ($< 30$M parameters), 'medium' ($\approx 50$M params) and 'large' ($\geq 80$M params) depending on their parametric size.

### A.2   Pre-training, augmentations and longer training

For the first three rows in 'Training Scheme' column in Table 1 we do 50 epoch adversarial training. For the specific cases of the 'Training Scheme' column, we have the following settings (the list is incremental):

- **random init., basic augment.** It has random initialized weights with RandomCrop as augmentation.

---

[2]https://github.com/facebookresearch/fvcore/tree/main/fvcore

- **+ strong clean pre-training.** Adversarial training is initialized with ImageNet-1k pre-trained weights from `timm` with RandomCrop as augmentation.

- **+ heavy augmentations.** It uses the augmentations similar to [34], including CutMix [65], MixUp [67] with strength 0.8, RandAugment [12] with two layers, magnitude 9 and a random probability of 0.5. Label smoothing is set to 0.1, Exponential Moving Average (EMA) with factor 0.999 and random erasing with probability 0.25. Unlike [34], we do not use any stochastic depth.

- **50 → 300 epochs.** We adversarially train for 300 epochs instead of 50 and peak LR is achieved at epoch 20.

The two Isotropic ConvNeXt models in Table 1 use the pre-trained weights from the original repository[3] as these are not available on `timm`.

### A.3    Scaling-up and comparison to SOTA models

For the results in Sec. 5, all our runs were initialized with ImageNet-1k pre-trained weights and use the heavy augmentations protocol from the previous section. Adversarial training was done with 2-step APGD for either for 50, 250 or 300 epochs (as listed in the 'Epochs' column in Table 2). For all 'ours' models in Table 2 in 'small' category the effective batch size was 1392, for 'medium' it was 1024. In 'large' category, all ViT-B and ConvNeXt-B models have an effective batch size of 864 and 756 respectively. Unlike for 'small' models, in 'medium' and 'large' category the checkpoint selection is done using 10-step $APGD_{CE}$ attack (first attack in AutoAttack), since 1-step $APGD_{CE}$ returns several checkpoints in a very small numerical range of robustness values.

### A.4    Increasing test-time resolution

When using image resolution other than 224x224, we keep the same pre-processing, only resolution is changed. For final resolution NS (new size) and scale-ratio (SR) 0.875 (same as during training), the images are resized with bicubic interpolation to NS/SR and then a center crop to NS is done. Note that we do not fine-tune the models with higher resolution images.

### A.5    Fine-tuning to larger radius

Fine-tuning to larger radii (Sec. 6) uses our ViT-S + ConvStem, ConvNeXt-T + ConvStem and 50 epochs ConvNeXt-B + ConvStem models from Table 2 as initialization. We employ the standard setting with heavy augmentations and adversarially train for 25 epochs at the $\ell_\infty$ radius of $8/255$ with a peak LR of 1e-4 attained at epoch 5.

## B    Design Choices for ConvStem

In this section, we give a detailed account of how ConvStem can be designed for different architectures and how it affects model size (FLOPs, parameters) and robustness. All the ConvStem configurations discussed below had the non-ConvStem part of the model initialized with pre-trained ImageNet-1k weights and the ConvStem with random initialization. This setup was chosen to manage the computation load. Afterwards, for the best configurations selected, we clean trained the model with ConvStem before training them adversarially.

### B.1    A closer look at the convolutional stem

We here investigate which elements of the convolutional stem are most relevant for better robustness of ConvNeXt-T to unseen threat models. Standard ConvNeXt has a 4x4 convolution in the initial stem (see Fig. 2, right column) that downsamples the input resolution by 4 with non-overlapping patches. This means any possible ConvStem design is limited to a maximum of two downsampling operations. Further, to constrain the increase in parameters and FLOPs we limit our choices to the configurations in Table 7, where we also report the corresponding results. We start off with the standard PatchStem from [34] ($Conv^{4,4}$, L), with L indicating LayerNorm and G a GELU layer. In ($Conv^{7,4}$, L, G), we increase the kernel size from PatchStem which results in mixing of feature maps,

---

[3] https://github.com/facebookresearch/ConvNeXt

Table 7: **ConvStem design for ConvNeXt.** We progressively modify the stem of ConvNeXt and track its effect on the robustness in seen and unseen threat models. The stem design is represented by the sequence of convolutional layers (Conv$^{x,y}$, i.e. kernel size x and stride y), LayerNorm (L), and GELU (G).

| Stem design | FLOPs (G) | Adversarial Tr. w.r.t. $\ell_\infty$ | | | |
|---|---|---|---|---|---|
| | | clean | $\ell_\infty$ | $\ell_2$ | $\ell_1$ |
| Conv$^{4,4}$, L [34] | 4.49 | 71.0 | 46.5 | 38.1 | 14.9 |
| Conv$^{7,4}$, L, G | 4.50 | 69.8 | 46.6 | 38.4 | 17.6 |
| Conv$^{7,4}$, L, G, Conv$^{3,1}$, L, G | 4.63 | 67.5 | 43.6 | 37.9 | 15.7 |
| Conv$^{7,2}$, L, G, Conv$^{3,2}$, L, G | 4.66 | 69.6 | 47.6 | 38.8 | 18.7 |
| Conv$^{5,2}$, L, G, Conv$^{3,2}$, L, G | 4.62 | 70.2 | 47.4 | 44.2 | 21.2 |
| Conv$^{3,2}$, L, G, Conv$^{3,2}$, L, G | 4.59 | 69.5 | 46.2 | 47.8 | 27.7 |

but this change does not yield any dividends in any of the metrics. Now we add another layer in our convolution stem. From (Conv$^{7,2}$, L,G - Conv$^{3,2}$, L, G) we keep the kernel size in the first layer 7 and vary the stride in second layer depending on the amount of downsampling required to match the standard ConvNeXt. For all these settings we do not see any improvement in $\ell_2$ and $\ell_1$ robustness but there are some gains in $\ell_\infty$ robustness. This increase in $\ell_\infty$ can be attributed to an increase in FLOPs. However, when trained with a good clean trained models as initialization, all these setting have a very similar $\ell_\infty$ robustness. As next step, we reduce the kernel size in the first layer to 5 in configuration (Conv$^{5,2}$, L,G - Conv$^{3,2}$, L,G), which leads to gains across all metrics, and reducing the kernel size to 3 yields further improvements in unseen threat models ($\ell_2$ and $\ell_1$). Since these last two configurations look very promising, we trained clean models with such architectures to a similar level of clean performance and used them as initialization for adversarial training: we observed negligible difference across all metrics. Given (Conv$^{3,2}$, L,G - Conv$^{3,2}$, L,G) has lower FLOPs of the two we select this configuration as main design for ConvStem. From Table 14, this configuration only leads to 2.9% and 0.1% increase in FLOPs and parameters respectively.

Additionally, for ConvNeXt-B we use the same stem as ConvNeXt-T and it slightly improved $\ell_2$ and $\ell_1$ robustness without improving $\ell_\infty$ robustness. We also found that adding an extra layer to this stem improved the $\ell_2$ and $\ell_1$ numbers (see Table 8) with a negligible increase in FLOPs and parameters. As this is a large model, an ablation similar to the one for ConvNeXt-T was not feasible. In the end, we use $C_2$ (from Table 8) as ConvStem for our ConvNeXt-B models. From Table 14, this ConvStem increases the FLOPs by around 4% and the parameters by only 0.2%.

## B.2   ViT ConvStem

We here study various designs of ConvStem on ViT-S (to keep the computational load low). We recall that for ViT the standard PatchStem generates 16x16 patches, which means we have a higher degree of freedom as to how the ConvStem can be designed compared to ConvNeXt (while keeping the increase in FLOPs from the standard model minimal).

In Table 9, we start off with the ConvStem as proposed by [62]: this configuration ($C_1$) results in a 8.1% increase in FLOPs but the improvement in all metrics tested is quite significant. Now we limit the increase in FLOPs to this value and try several other configurations within this limit. Two

Table 8: **ConvStem configurations for ConvNeXt-B.** Each layer in configuration column is followed by LayerNorm and GELU activation. * indicates layers with stride 2 (one downsampling operation), and N (= 64) is the number of output channels of the convolution layer. The increase/decrease in numbers from the PatchStem are shown in green/red respectively.

| Stem design | Adversarial Tr. w.r.t. $\ell_\infty$ | | | |
|---|---|---|---|---|
| | clean | $\ell_\infty$ | $\ell_2$ | $\ell_1$ |
| $P_1$: PatchStem | 75.6 | 54.3 | 48.5 | 23.7 |
| $C_1$: N*-2N* | 75.2 -0.4 | 54.2 -0.1 | 48.9 +0.4 | 24.0 +0.3 |
| $C_2$: N*-1.5N*-2N | 75.3 -0.4 | 54.4 +0.1 | 50.1 +1.6 | 24.7 +1.0 |

Table 9: **ViT-S ConvStem ablation.** N, 2N, 4N, 8N are the number of output channels in each convolution layer, where N=48, similar to that of [62]. Each layer is followed by LayerNorm and GELU activation. *: kernel-size 3 and stride 2 (one donwsampling in the layer), **: kernel-size 7 and stride 4 (double donwsampling in the layer).

| Stem design | FLOPs (G) | Adversarial Training | | | |
| --- | --- | --- | --- | --- | --- |
| | | clean | $\ell_\infty$ | $\ell_2$ | $\ell_1$ |
| $P_1$: PatchStem | 4.61 | 65.1 | 39.2 | 37.4 | 16.5 |
| $C_1$: N*-2N*-4N*-8N* | 4.99 | 68.0 | 43.3 | 46.3 | 27.0 |
| $C_2$: N*-N*-N*-8N* | 4.71 | 64.8 | 41.1 | 45.3 | 28.3 |
| $C_3$: N**-8N** | 4.78 | 64.5 | 40.3 | 46.2 | 32.3 |
| $C_4$: N**-2N*-4N*-8N | 4.84 | 64.7 | 40.4 | 46.4 | 32.6 |
| $C_5$: N**-2N**-4N-8N | 4.81 | 64.6 | 39.3 | 45.8 | 33.2 |

ways to change the configuration are most trivially feasible here: 1) reducing the number of output channels in each convolution layer and 2) changing the level of downsampling done in each layer. As the combination of these two changes most readily leads to the change in FLOPs, we only apply these two kind of changes in our ablation study. Starting from $C_1$, we first reduce the number of output channels in each layer, since the input to the transformer block is 8N (the last layer's output channels cannot be changed). Both $C_2$ and $C_3$ lead to significant reduction in $\ell_\infty$ numbers, hence we shift towards changing the amount of downsampling in each layer. Moving more downsampling to earlier layers further reduces $\ell_\infty$ robustness compared to $C_1$. This small ablation leads us to fix the ConvStem for ViT-S to configuration $C_1$. From Table 14, the increase in FLOPs is only 8.1% and parameters increase by 3.3%.

We use the same ConvStem as ViT-S with final layers output channels scaled to $16N$ to meet the input channel for transformer block in ViT-B. This only increases the FLOPs by 2% and parameters by a mere 0.6%. Similarly for our ViT-M, we again use the same ConvStem as ViT-S with last layer's output size set to 512.

**Improvements in robustness with ConvStem.** Our experiments throughout the paper, validate that a minimal change in the network architectures, adding the ConvStem provides consistent improvements in robustness of the adversarially trained models. Fig. 4 summarises this and shows how across network types the classifiers with ConvStem achieve better robustness in both seen ($\ell_\infty$) and unseen ($\ell_2$) threat models.

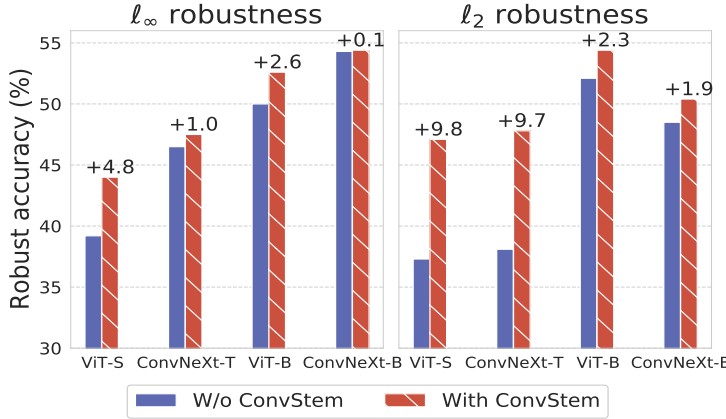

Figure 4: **ConvStem yields consistent improvements across architectures, model sizes in seen and unseen robust accuracy.** We show for 50 epochs of AT with pre-training and heavy augmentation, that adding ConvStem yields for ViT and ConvNeXt robustness improvements both in seen $\ell_\infty$ (left subplot) and the unseen $\ell_2$ (right subplot) threat models.

Table 10: **Clean accuracy of standard trained ConvStem models.** For all the main models in this work, we show their clean performance on RobustBench 5k validation set. The difference to non-ConvStem models with `timm` initialization is shown in red.

| Architecture | Epochs | Clean acc. | |
|---|---|---|---|
| ConvNeXt-T + ConvStem | 100 | 80.9 | -0.5 |
| Isotropic ConvNeXt + ConvStem | 100 | 79.2 | -0.5 |
| ViT-S + ConvStem | 100 | 79.3 | -1.0 |
| ViT-M + ConvStem | 50 | 81.8 | -0.6 |
| ConvNeXt-S + ConvStem | 50 | 81.6 | -1.0 |
| ViT-B + ConvStem | 50 | 81.6 | -1.1 |
| ConvNeXt-B + ConvStem | 50 | 82.6 | -0.8 |

### B.3 Clean performance of ConvStem models

Our ConvStem models from Sec. 4 onward always use a standard classifier as initialization for adversarial training. This clean training was done either for 50 or 100 epochs depending on the size of the model, see Table 10. The non-ConvStem part of the model (ConvNeXt stages and all transformer blocks) were initialized with `timm` ImageNet-1k pre-trained weights and the ConvStem was randomly initialized. In Table 10, we show that for all models we were able to reach within 1% of clean performance of standard models on the RobustBench validation set. Certainly, initialization of non-ConvStem part helps in this fast and efficient learning. The good clean accuracy via short training also corroborates the improved training hypothesis of [62] when using ConvStem instead of a PatchStem without much hyperparameter tuning.

Table 11: **Comparison to [14] with respect to training time.** Both our ConvNeXt and the ConvNeXt + ConvStem attain higher robustness in the $\ell_\infty$ threat model than the ConvNeXt of [14] while using less forward/backward passes per image (see discussion at the beginning of Sec. 5).

| Model | Epochs | Attack Steps | Passes/Image | $\ell_\infty$ |
|---|---|---|---|---|
| ConvNeXt-T [14] | 110 | 1 | 220 | 44.4 |
| ConvNeXt-T | 50 | 2 | 150 | 46.5 |
| ConvNeXt-T + ConvStem | 50 | 1 | 100 | 46.4 |
| ConvNeXt-T + ConvStem | 50 | 2 | 150 | 47.5 |
| ConvNeXt-T + ConvStem | 300 | 2 | 900 | 49.5 |
| ConvNeXt-T + ConvStem | 300 | 3 | 1200 | 50.2 |

## C  Design choices, cost/size comparisons and new architectures

In this section, we answer *(i)* How the total training cost of our training regime compares to existing works *(ii)* What is the effect of using different optimizer for adversarial training *(iii)* Do the improvements as seen by adding ConvStem hold for the most recent architectures *(iv)* How does the model size change on adding ConvStem.

### C.1 Training cost comparison

Table 11 shows the robustness in the seen $\ell_\infty$ threat model of ConvNeXt-T model trained with strong clean pre-trained initialization and heavy augmentations when varying training cost. In particular, the training scheme cost is approximated by the number of passes of the network for each training image (counting the number of forward+backward passes per image combines the number of attack steps with training epochs used). First, we show that our ConvNeXt-T with 150 passes/images attains 2.1% higher robustness (46.1% vs 44.4%) than the same model trained with 220 passes/image by [14]. Moreover, adding the ConvStem achieves a similar result with only 100 passes/image (third row), and 47.5% robust accuracy with 150 passes/image (fourth row). Finally, increasing the number of epochs and training steps, which incurs in more expensive training, can further improve robustness.

Table 12: **PGD vs APGD.** Comparing the difference in robustness on using PGD instead of APGD (used in this work) trained for 2 steps each on a ConvNeXt-T. We evaluate clean performance and robustness.

| Optimizer | clean | $\ell_\infty$ | $\ell_2$ | $\ell_1$ |
|---|---|---|---|---|
| PGD | 72.8 | 45.4 | 39.1 | 16.5 |
| APGD | 71.0 | 46.5 | 38.1 | 14.9 |

Table 13: **Effect of replacing ConvStem with PatchStem in InternImage.** We train the InternImage (original with ConvStem) and InternImage with PatchStem (PS) for 50 epochs in our heavy augmentation setup.

| Architecture | clean | $\ell_\infty$ | $\ell_2$ | $\ell_1$ |
|---|---|---|---|---|
| InternImage-T | 72.9 | 47.0 | 48.5 | 25.9 |
| InternImage-T + PS | 72.7 | 47.0 | 47.2 | 24.5 |

## C.2 APGD vs. PGD training

The optimizer for adversarial training is also a design choice to be considered while training robust models. Prior works [14, 46] mostly use either PGD [36] or APGD [8]. We use APGD since it has been shown to generally outperform PGD and does not need parameter tuning when changing radius or attack steps [8]. Additionally, in Table 12 we train a ConvNeXt-T with 2-step PGD (clean pre-training, heavy augmentations, 50 epochs), comparable to the classifier using 2-step APGD from Tables 1 and 2. In this regime APGD yields higher robustness in the seen ($\ell_\infty$) threat model which is our main goal, although with a small loss in clean accuracy.

## C.3 Testing more architectures

We further test the effect of different stem designs on InternImage [57], a modern architecture that uses deformable convolutions and has a ConvStem as its stem. In Table 13, we replace the ConvStem with a PatchStem and evaluate robustness for the seen ($\ell_\infty$) and unseen ($\ell_2, \ell_1$) threat models. The structure of the PatchStem (PS) is the same as for ConvNeXt. For the PS model, we initialize the PS randomly and do clean pre-training for 50 epochs to reach 82.8% clean accuracy (the original InternImage with ConvStem has 83.5%). Similar to ViT and ConvNeXt, we see that robustness for the unseen threat models goes down when replacing the ConvStem with the PatchStem. The models were adversarially (2-step APGD) trained for 50 epochs with ImageNet-1k clean trained model[4] as initialization and with the heavy augmentation regime, while other parameter setting were kept same as heavy augmentations in App. A.2.

Table 14: **FLOP and parameter count**. Here we list the FLOPS (Giga), Parameters (Millions) of all the main models and their ConvStem versions. The percentage increase in values on adding ConvStem are shown in red.

| Architecture | FLOPs (G) | | Params (M) | |
|---|---|---|---|---|
| ConvNext-T | 4.47 | | 28.59 | |
| + ConvStem | 4.60 | +2.9% | 28.63 | +0.1% |
| Isotropic ConvNeXt-S | 4.29 | | 22.31 | |
| + ConvStem | 4.67 | +8.8% | 23.04 | +3.3% |
| ViT-S-16 | 4.61 | | 22.05 | |
| + ConvStem | 4.99 | +8.1% | 22.78 | +3.3% |
| ViT-M-16 | 8.01 | | 38.85 | |
| + ConvStem | 8.38 | +4.7% | 39.5 | +1.7% |
| ConvNeXt-S | 8.70 | | 50.10 | |
| + ConvStem | 8.79 | +1.7% | 50.33 | +0.5% |
| ViT-B-16 | 17.58 | | 86.57 | |
| + ConvStem | 17.93 | +2.0% | 87.14 | +0.6% |
| ConvNeXt-B | 15.38 | | 88.59 | |
| + ConvStem | 15.97 | +3.8% | 88.75 | +0.2% |

---

[4] https://github.com/OpenGVLab/InternImage

Table 15: **Effect of using 21k-pre-trained models.** Across the two best models discovered we see that there is no improvement in the main seen threat on initializing with ImageNet-21k pretrained weights from `timm`. The increase/decrease from ImageNet-1k counterparts of the same models are shown in green/red colors respectively.

| Architecture | Epochs | clean | | $\ell_\infty$ | |
|---|---|---|---|---|---|
| ConvNeXt-T | 50 | 71.6 | +0.6 | 46.7 | +0.2 |
| ViT-B-16 | 50 | 73.5 | +0.2 | 49.6 | -0.4 |
| ConvNeXt-T | 300 | 72.2 | +0.2 | 48.3 | -0.3 |

## C.4  Increase in paramaters/FLOPs

In Table 14, we list the number of parameters and FLOPs of all models, with the percentage increase on adding the ConvStem shown in red: for all models adding ConvStem results in a marginal amount of extra parameters. For FLOPs, smaller ViTs have a bigger increase as compared to ConvNeXt. This can be attributed to the bigger size of ConvStem we use for ViT as we need to downsample before the first transformer block more severely than for ConvNeXt.

# D  More on Robust ImageNet Models

As the main aim of our work is to study and improve robustness of models trained for ImageNet to seen and unseen threat models, in this section we consider ablations (effect of clean pre-traininig, data augmentations, weight-decay and label smoothing) related to this. Moreover, we provide more details about using robust ImageNet for fine-tuning to other datasets.

## D.1  Using even better pre-training

The most robust models in our work use ImageNet-1k pre-trained weights of the respective architectures from `timm`. A natural hypothesis is that a better pre-trained model, namely an ImageNet-21k pre-trained model fine-tuned on ImageNet-1k, might lead to higher robustness. `timm` makes available for architectures considered in this work (ViT, ConvNeXt) 21k-pretrained, 1k-fine-tuned weights as well. In Table 15, we see the effect of using 21k-pretrained weights as initialization for adversarial training. The increase and decrease in clean accuracy and $\ell_\infty$ robust accuracy as compared to the 1k-pre-trained initialization of the same models is shown in green and red respectively.

The ConvNeXt-T trained for 50 epochs shows a small increase in $\ell_\infty$, while the one trained for 300 epochs gets slightly lower robustness. For ViT-B with 50 epochs we see a drop of 0.4% in robust accuracy although clean accuracy goes up by 0.2% (which is intuitively correct as adversarial training starts from a better point with 21k-pre-trained weights). Overall we did not see any sign of gains in robustness when using a model pre-trained on a larger dataset. This finding is line with what [38] witnessed when using ImageNet-1k and 21k as initialization while training robust models for CIFAR-10.

Table 16: **Ablation for data augmentations.** We show how varying levels of data augmentations influences robustness across all threat models in our training paradigm. The best performing augmentation setup for each model per perturbation is in **bold**. All models are trained for 50 epochs. Results for basic and heavy augmentation are taken from Table. 1.

| Augmentation | ViT-S + ConvStem | | | | ConvNeXt-T + ConvStem | | | |
|---|---|---|---|---|---|---|---|---|
| | clean | $\ell_\infty$ | $\ell_2$ | $\ell_1$ | clean | $\ell_\infty$ | $\ell_2$ | $\ell_1$ |
| basic | **71.2** | **44.3** | 47.1 | 23.2 | 69.1 | 42.2 | 42.4 | 19.7 |
| 3-Aug [14] | 70.1 | 43.6 | 46.1 | 23.4 | **71.2** | 45.3 | 45.6 | 19.6 |
| heavy | 69.9 | 44.0 | **47.1** | **25.9** | 69.1 | **47.5** | **47.8** | **24.6** |

## D.2 Ablating data augmentation components.

A complete ablation study of which augmentations are useful for adversarial training on ImageNet is beyond the scope of this paper. The results of Table 1 suggest that heavy augmentations works better than the basic augmentation (random crop). In this section we test additionally the intermediate augmentation suggested in [14] called 3-aug (basic + radom horizontal flip + color-jitter), in particular we are interested in the performance of the ConvStem models and the generalization beyond the $\ell_\infty$-threat model used for training. In Table 16 we show for two models, ViT-S + ConvStem and ConvNeXt-T + ConvStem, each trained for 50 epochs the results of basic, 3-Aug [14] and heavy augmentation. Note, that the clean pre-trained model was trained with the heavy augmentations protocol. We can see that using heavy augmentation yields the best robustness across all threat models for the ConvNeXt-T+ConvStem ($+2.2\%$ in $\ell_\infty$-robustness) while having slightly worse clean accuracy ($-1.1\%$). For the ViT-S+ConvStem the results are mixed, basic augmentation performs best, heavy augmentation is second and 3-Aug the worst of all three. However, the gap between basic and heavy augmentation is marginal and thus we stick to heavy augmentation throughout the paper. It is an interesting open question if there exist intermediate or completely different augmentation schemes which would yield improved robustness for ImageNet. A more detailed analysis for the effects of data augmentation on adversarial robustness for small resolution datasets like CIFAR-10 or SVHN can be found in [31].

## D.3 Effect of weight decay and label smoothing on adversarial robustness

For our training scheme which already yields strong results (50 epochs of adversarial training with strong pre-training and strong augmentation), we test the effect of varying the weight decay and label smoothing parameters. We check on a logarithmic scale other weight decay values than our default value 0.05. In Table 17, we see that for both ViT and ConvNeXt neither smaller nor larger weight decay yields any improvement. Moreover, we observe that the label smoothing parameter has, compared to weight decay, less influence on the final robustness. For the ViT we get a small improvement of $0.2\%$ when using label smoothing coefficient 0.2 compared to our default value of 0.1, while for the ConvNeXt the results get $0.2\%$ worse for label smoothing coefficient 0.2. Then, we decided to keep a label smoothing coefficient of 0.1. Finally, we note that a weight decay of 0.05 and label smoothing 0.1 are the values used for clean training of the ConvNeXt in [34].

## D.4 Clean and robust accuracy trade-off

Fig. 5 shows the clean vs robust accuracy trade-off for ImageNet models. In the 'small' size category, our models, i.e. ViT-S-ConvStem (black circle) and ConvNeXt-T-ConvStem (red circle), have the best clean accuracy except for XCiT-S of [14], which however has more than 7% worse robust accuracy. In 'medium' size category our models were only trained for 50 epochs (see Table 2) but outperform all SOTA models in terms of robust accuracy. On top of that, our ConvNeXt-S-ConvStem has both the best clean and robust accuracy in this size range. For 'large' sized models only the ViT-B of [44] is in a comparable range. Their model has around 0.6% higher clean accuracy than our ConvNeXt-B-ConvStem but 2.6% lower robust accuracy. In general, our models mostly show a better clean vs robust accuracy trade-off as compared to existing models in literature.

Table 17: **Effect of weight decay and label smoothing coefficient.** Varying the weight decay and label smoothing coefficient around our default values (marked below with a *), which are the ones suggested for clean training in [34], does not lead to a consistent improvement in $\ell_\infty$ robustness.

| Architecture | Weight decay | | | Label smoothing | | |
|---|---|---|---|---|---|---|
| | 0.5 | 0.05* | 0.005 | 0.0 | 0.1* | 0.2 |
| ViT-S + ConvStem | 37.2 | 44.0 | 44.0 | 44.0 | 44.0 | 44.2 |
| ConvNeXt-T + ConvStem | 27.5 | 47.5 | 47.2 | 47.4 | 47.5 | 47.3 |

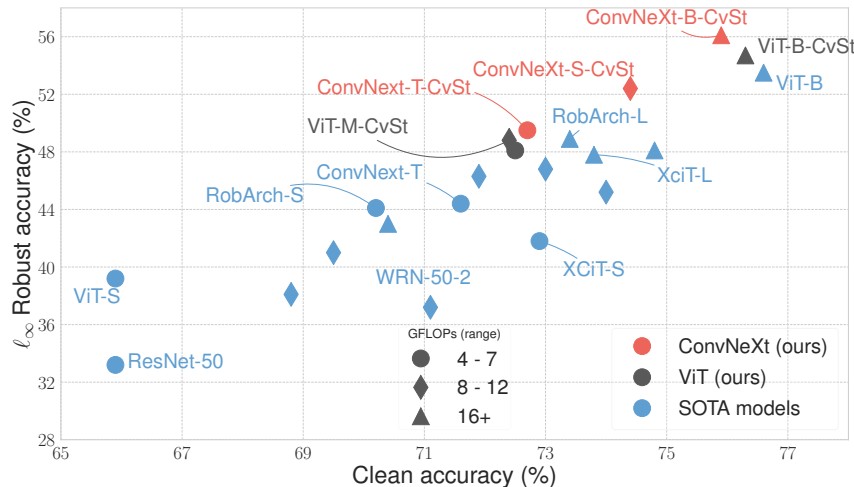

Figure 5: **AutoAttack $\ell_\infty$-Robust accuracy ($\epsilon = 4/255$) vs Clean accuracy on ImageNet across different architectures.** All our models are better than their SOTA counterparts in robust accuracy with only negligible difference in clean accuracy, if at all. ConvStem is written as CvSt.

## D.5    Increased image resolution

We further discuss the effect of varying the test-time resolution for $\ell_\infty$ robust models. All evaluations in the following are done with APGD$_{\text{T-DLR}}$ (40 iteration, 3 target classes) on 1000 test points to limit computational cost, as increasing resolution significantly impacts the number of FLOPs.

**Larger models.** Similar to 'small' models, 'large' models trained for 50 epochs also show improvements in both clean accuracy and $\ell_\infty$ robustness, see Fig. 6. This phenomenon is seen for both ViT and ConvNeXt with and without ConvStem, with peak robustness at resolution 256x256 for all classifiers.

$\ell_2$ **radius scales with resolution.** Since $\ell_\infty$ bounds are applied element-wise, increasing the resolution, i.e. the input dimension, does not change how much each input component can be modified. Thus one can argue that for fixed radius, the $\ell_\infty$-threat model is for increasing resolution/dimension at least as strong as the original one. In fact, as each dimension can be manipulated independently one can argue that increasing dimension makes the $\ell_\infty$-threat model more powerful. However, with a bound $\epsilon_2$ w.r.t. $\ell_2$ and input dimension $d$, the average element-wise perturbation is $\epsilon_2/\sqrt{d}$. If one keeps the $\epsilon_2$ constant for increasing dimension, it seems clear that the threat model becomes weaker.

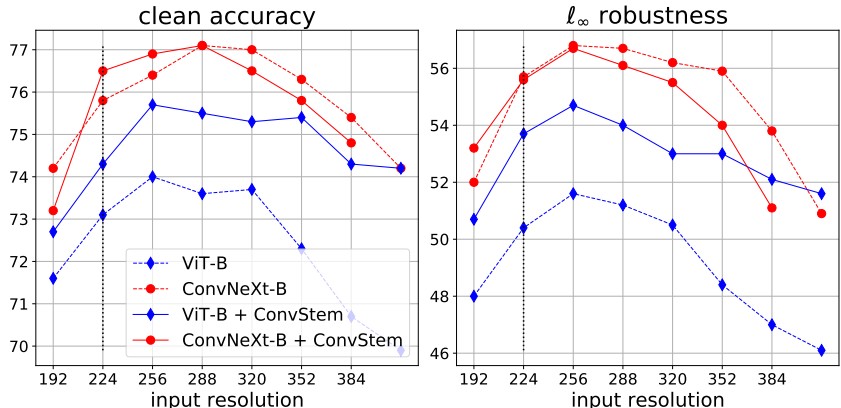

Figure 6: **(Robust) accuracy improves with increasing test-time image resolution.** Evaluation of clean (left) and $\ell_\infty$-robust accuracy (right) for the 50 epoch 'large' models from Table 2 when varying the input resolution at test-time.

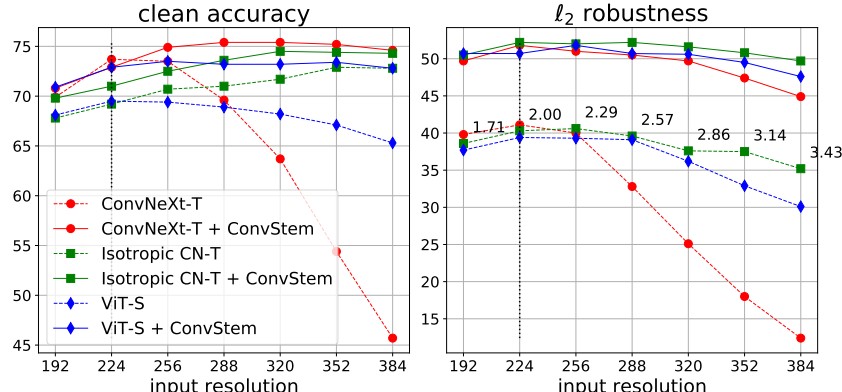

Figure 7: $\ell_2$-**(robust) accuracy for varying test-time image resolution.** Evaluation of clean (left) and $l_2$-robust accuracy (right) for the top models from Table 1 when varying the input resolution at test-time. The radius is $\epsilon_2 = 2$ for image resolution 224 and rescaled as $\epsilon_2 = 2\frac{d}{224}$ for varying $d$ (this means that the perturbation budget per pixel is constant), and can be read in the figure on the right.

In order to keep the perturbation budget per dimension constant we thus increase $\epsilon_2$ as a function of $\sqrt{d}$. Note however that when increasing $\epsilon_2$ the maximal perturbation per pixel or the number of pixels which can be maximally perturbed grows as well. Thus one can argue that the threat model becomes stronger when scaling with $\sqrt{d}$ when increasing $d$. In summary, we see that the comparison across image resolutions for $\ell_2$ (and similarly for $\ell_1$) is non-trivial. We decided to consider the scenario where the $\ell_2$-treat model is at least not becoming less powerful with increasing dimension and thus increase $\epsilon_2$ with $\sqrt{d}$. The resulting $\ell_2$-robust accuracies for varying resolution can be found in Fig. 7, where we see that the ConvStem- models are very stable in robust accuracy even for much higher resolution.

### D.6 Fine-tuning to other datasets

We study whether it is possible to transfer adversarial robustness from ImageNet to other datasets via fine-tuning our robust models. We consider datasets with images of resolution either similar to ImageNet, i.e. Flowers-102 [39], or smaller like CIFAR-10 and CIFAR-100 [29], and fine-tune the $\epsilon_\infty = 8/255$ robust models from Table 4. We fine-tune with 2 steps (Flowers-102) or 10 steps (CIFAR-10, CIFAR-100) adversarial training at $\epsilon = 8/255$ on all target datasets, with RandAug as augmentation, for 20 epochs, as in [14], with learning rate (LR) warm-up of 2 epochs: for Flowers-102 we set peak LR to 4e-3, a weight decay of 5e-3 and batch size 30, for CIFAR-100 peak LR is 2e-4, weight decay 0 and batch size 1024, for CIFAR-100 peak LR is 2e-4, weight decay 5e-3 and batch size 256. To adapt the networks to the new datasets, we replace the final linear classifier with one suitable for the new number of classes, and for smaller resolution inputs, we change from ConvNeXt + ConvStem the strides from 2 to 1 in the convolutional layers of the stem and in the first downsampling block in ConvNeXt. The robust accuracy is computed on the full test sets with AutoAttack in the resolution of the dataset to which we finetune, e.g. $32 \times 32$ for CIFAR-10/100.

Table 18: **Specific configurations of pre-trained models we use from** `timm`**.** As all models have a lot of pre-trained weights available on `timm`, we explicitly list the configuration (also includes pre-trained weight path) of the specific architectures.

| Architecture | Configuration |
|---|---|
| ConvNeXt-T | `convnext_tiny.fb_in1k` |
| ConvNeXt-S | `convnext_small.fb_in1k` |
| ConvNeXt-B | `convnext_base.fb_in1k` |
| ViT-S | `deit_small_patch16_224` |
| ViT-M | `deit3_medium_patch16_224` |
| ViT-B | `vit_base_patch16_224.augreg_in1k` |

### D.7 More pre-training details

In this section we give some additional details related to our models. As we heavily rely on pre-training in our work, the origin of pre-trained models on `timm` are listed in Table 18. Most ViTs and ConvNeXts available at `timm` have different versions (patch sizes, resolutions) and even pre-training (ImageNet-21k, -12k, -1k). Among those we use ViTs with patch size 16 trained at resolution of 224 on ImageNet-1k only. Similarly, for ConvNeXt we use the ones trained only with ImageNet-1k and resolution 224. For ViT-M, we use pre-trained weights of a DeiT-3 model from [51], because ViT-M was not available in our desired configuration of ImageNet-1k only pre-training. Note that ViT and DeiTs have the same architecture and differ only in the training scheme. All the clean pre-trained models use heavy augmentations protocol except ViT-M (DeiT-3-M) which uses only 3 augmentations. Finally, for Isotropic ConvNeXt-S we use the weights of `convnext_iso_small_1k_224_ema` from the original repository.

