# OpenReview forum: "Revisiting Adversarial Training for ImageNet: Architectures, Training and Generalization across Threat Models"
_NeurIPS.cc/2023/Conference — NeurIPS 2023 poster_

### Official Review · Reviewer_qHdB · 2023-06-14

**Soundness:** 2 fair
**Presentation:** 2 fair
**Contribution:** 3 good
**Rating:** 5
**Confidence:** 4

**Summary:**

This work investigates the effect of a set of architectural, training and inference techniques on adversarial training (AT) on ImageNet. AT has been less studied on ImageNet compared to CIFAR10 due to the much computational cost, but the behavior of AT on large-scale realistic dataset is undoubtedly an important topic, so I appreciate a lot the authors’ efforts on this direction. In my understanding, the key takeaway techniques suggested by the paper for improving robustness on ImageNet are ConvStem, pre-training followed by heavy data augmentation, training longer and increasing resolution at inference time. To me, the first and fourth points are novel, while the second and third points are somehow expected. My biggest concern is the lack of explanation behind these empirical observations like why ConvStem works and the generalization abilities of these tricks particularly across various architectures and AT methods. The analysis and explanation are particularly important because none of these techniques is invented in this work, i.e., they are originally proposed by the previous works. Please see the details below. I will be happy to raise my scores if the above concerns are clarified or addressed.

**Strengths:**

1. Adversarial training on ImageNet is less studied and indeed an important direction for moving forward for the community

2. Identifying a good receipt of AT on ImageNet can have a great potential impact on the future works, e.g., adopted as a standard training setting for benchmarking various AT methods.

3. Robustness improvement of ConvStem on some architectures is significant

4. The influence of test-time image resolution is interesting.

**Weaknesses:**

1. ConvStem seems to only work well with Isotropic architectures but work less well or even not work with ConvNeXt regarding the robustness evaluated against the same threat model as training’s, i.e., L inf. For example, it improves robustness by 0.5% on ConvNeXt-T with basic training setting (Table 1) and by 0.04% on ConvNeXt-B (Table 2). There is almost no improvement on ConvNeXt-B. Even worse, the accuracy seems to be sacrificed for improved robustness on ConvNeXt-B and in some settings of ConvNeXt-T like “+ heavy augmentation”. Therefore, I doubt the generalizability of ConvStem across the different architectures. It will be interesting to test if ConvStem can benefit robustness for other non-isotropic CNN-based architectures such as InternImage [1] which also uses ConvStem.

2. I find almost no analysis about why ConvStem works well on Isotropic architectures and less well on others, why pre-training enables the use of heavy data augmentation, and why training longer works in this context but not in [2]. This is important for us to understand the accounts for the performance boost and to judge the generalization ability of the method, especially when the suggested techniques are originally proposed in the previous works.

3. A major benefit of ConvStem is the robustness towards the unseen threat models like L1 and L2. However, the discussion of and the comparison with the related works developed for this purpose (trained on L inf to defend L1 and L2) are missing, so it is hard for me to judge how significant the reported improvement is and if the improvement still holds when combined with other techniques.

4. It is also unclear whether ConvStem can improve the SOTA AT methods like how it works with the baseline AT, or if not, whether ConvStem+baseline AT can beat the SOTA AT methods. All tested AT methods are APGD AT with only different steps: 1, 2, 3.

[1] InternImage: Exploring Large-Scale Vision Foundation Models With Deformable Convolutions, CVPR 2023

[2] When Adversarial Training Meets Vision Transformers: Recipes from Training to Architecture, NeurIPS 2022

**Questions:**

1. L43: The claim of SOTA adversarial robustness on ImageNet may not be right. [1] (arxiv version: 28 Feb 2023) reported a robustness of 56.8% for ConvNeXt-B, while the highest robustness for ConvNeXt-B+ConvStem reported in this work is 56.28%. On RobustBench,  you both submitted a result for ConvNeXt-L: 58.48% from [1] and 57.70% from yours with ConvStem. Therefore, 56.28%, even 57.3% with inference-time resolution technique, seems to be not the state-of-the-art robustness against AA on ImageNet.

2. Table 1: how many runs are repeated for the reported result? If more than one, why not report the standard deviation?

3. L107: why APGD-2, instead of more common attacker PGD-2?

4. L239-240: Have your single-step AT suffered from catastrophic overfitting? If not, did you apply any mitigation technique?

5. Table 2: some implementation details of several related works like “comprehensive” and “light” are missing, so that I can’t judge if the performance difference is caused by, e.g., ConvStem or something else. I will suggest the authors clearly discuss the technical difference between these training receipts and yours, e.g., organized in a table.

6. Section 4.2: how do you decide the specific data augmentation as the component of heavy data augmentation? Besides, it will be interesting to test how a recent data augmentation, IDBH, specific for AT works on ImageNet compared with others.

7. Section 6, test-time image resolution: have you tried to evaluate the robustness using larger epsilon? I guess the robustness improvement will disappear in the setting with larger potential adversarial vulnerability.

8. Table 2: the results of ViT-M and ConvNeXt-S without ConvStem are missing. Are there any reasons for this?

[1] A Comprehensive Study on Robustness of Image Classification Models: Benchmarking and Rethinking, Arxiv

[3] A Light Recipe to Train Robust Vision Transformers, SaTML.

[2] Data augmentation alone can improve adversarial training, ICLR 2023.

**Limitations:**

The authors discussed the limitation as “not covering more architectures and architectural components”. There are still some limitations about the generalizability of ConvStem across the architectures and the AT methods as listed in Weakness.

---

> ### Author Rebuttal · Authors · 2023-08-09
>
> We thank the reviewer for the detailed comments, and for their positive comments. Next we concisely, due to limited space, address the weaknesses pointed out in the review.
>
> **ConvStem seems to only work well with Isotropic…**
>
> We remark that for the best setup (“50->300 ep.”) in Table 1 the ConvStem provides significant improvements in $\ell_\infty$ robustness (+0.9%) also for the ConvNeXt, even with slightly better clean performance (+0.3%). For the base (B) models, the marginal improvement of 0.04% given by the ConvStem is for models trained for only 50 epochs. While for limited resources we could not train a ConvNeXt-B for 250 epochs (it takes ~12 days on 8 A100s), we note that our ConvNeXt-B+ConvStem models (with 2 or 3 attack steps) trained for 250 epochs outperform the ConvNeXt-B of [28] trained for 300 epochs with 3 attack steps by 0.3-0.4%: this suggests the positive impact of the ConvStem even on the large non-isotropic architectures.
> For a discussion about clean accuracy, please see the [reply to Reviewer bR1u](https://openreview.net/forum?id=Pbpk9jUzAi&noteId=roPu0dNG36).
> Since InternImage [1] already uses ConvStem, it is not clear to us how to show the effectiveness of ConvStem on it. Do you suggest replacing it with a PatchStem?
>
> **ConvStem more effective on Isotropic architectures**
>
> Please see the [global response](https://openreview.net/forum?id=Pbpk9jUzAi&noteId=3fBEqU4pBW).
>
>
> **Why pre-training enables heavy data augmentation**
>
> Adversarial training is a more challenging learning problem than clean training. Using heavy augmentations with a random initialization can make the model diverge (see Sec. 4.2 and [12]). When starting with the pre-trained model as initialization the robust loss is initially significantly smaller than for a random initialization and decreases also faster.
>
> **Why training longer works in this context but not in [2]**
>
> Mo et al. [2] experiment only with CIFAR-10 and Imagenette, i.e. smaller datasets than full ImageNet, where overfitting is a significant issue. For ImageNet this is a milder issue, although we observe that long training (300 ep.) without sufficiently heavy augmentations also leads to overfitting for ViT-S+ConvStem (see Table A1 above).
>
> **Related works about training on Linf to defend L1 and L2**
>
> We are not sure which works the reviewer refers to, but would be happy to discuss them if pointed to them. Tracking the robustness to unseen attacks is not standard, and we are not aware of techniques with the goal of training wrt $\ell_\infty$ to obtain robustness in $\ell_2$ and $\ell_1$.
>
> **Can ConvStem improve the SOTA AT?**
>
> We are not sure what the reviewer refers to as SOTA AT methods, but we compare our models with all existing SOTA classifiers, and improve upon them.
>
> **Claim of SOTA robustness**
>
> The 56.8% in Liu et al. [1] (ref. [28] in our paper) is computed on a different set of points as our results. All results in Table 2 use the subset of 5k validation points defined by RobustBench for a fair comparison. In this setup, the ConvNeXt-B of [1] attains 55.82% robust accuracy, which is 0.46% lower than our ConvNeXt-B+ConvStem (Table 2 has both models): as mentioned at L274-275, our model achieves this result even with a 20% shorter training time.
> For the ConvNeXt-L, we train ours for only 100 epochs and 2 attack steps, while [1] uses 300 epochs and 3 steps. Given the beneficial effect of longer training, especially for larger models, the two results are not directly comparable. Please note that with our computational resources (8xA100 GPUs), 100 epochs of ConvNeXt-L+ConvStem took 12 days to complete, which made scaling it up at least 3 times infeasible.
>
> **How many runs are repeated?**
>
> A single run due to the high computational cost.
>
> **APGD-2 instead of PGD-2**
>
> We use APGD since it has been shown to generally outperform PGD and does not need parameter tuning when changing radius or attack steps [8]. Additionally, we train a ConvNeXt-T with PGD-2 (clean pre-training, heavy augm., 50 ep.), comparable to the classifier using APGD-2 from Table 1. In this regime APGD-2 yields higher robustness, which is our main goal, although with a small loss in clean accuracy (Table A2 in the attached file).
>
> **Catastrophic overfitting?**
>
> Similar to the observations of [Andriushchenko & Flammarion, [2020]](https://arxiv.org/abs/2007.02617) for small (2/255, 4/255) radii on ImageNet, we did not experience catastrophic overfitting in any setup, including single step training, even without any mitigation technique.
>
> **Experimental details of other papers**
>
> Table 2 includes the main parameters (epochs, attack steps), and additional details can be found in the original papers. Please note that we mostly follow the training setup of Liu et al. [30] (which is similar for other works too), without adjusting it to our classifiers. We will add a summary table with the parameters from each paper.
>
> **Choice of heavy augmentations, testing IDBH**
>
> The heavy data augmentation simply follows the setup used in Liu et al. [30], which is common, with little variations, to other works e.g. [29, 40]. IDBH has not been tested on ImageNet but only smaller datasets, and extending it is out of the scope of our work.
>
> **Test-time resolution at larger radii**
>
> Table A3 (attached file) reports robust accuracy varying perturbation radii and image resolutions: even at the larger 6/255 and 8/255 higher test-time resolution (288 and 256 respectively, 224 is used for training) improves robustness. However, as anticipated by the reviewer, we see a diminishing advantage from increasing resolution for larger radii.
>
> **Medium models without ConvStem**
>
> We prioritized the model sizes with most competitors. Table A4 (attached file) now illustrates that the ConvStem provides improvements in almost all metrics for medium models too, in particular for the $\ell_\infty$-robustness of ViT-M (+1.6%), and the robustness to $\ell_2$ (+7.1%) and $\ell_1$ (+6.1%) for ConvNeXt-S.

---

> > ### Comment · Reviewer_qHdB · 2023-08-11
> >
> > Thanks for the authors' detailed response and added experiments. Some of my major concerns are still not addressed.
> >
> > > Generalization of ConvStem across different model architectures (mainly non-isotropic architectures due to marginal improvement) on improving robustness against seen attack Linf.
> >
> > For non-isotropic architectures, this work only tests ConvNeXt family with T, S, B three sizes, and two sizes of them S and B show a <= 0.1% robustness improvement. The authors claim that this is due to the insufficient epochs for training. However, the given argument of ConvNeXt-B+ConvStem vs. ConvNeXt-B [28] can be unfair, because they adopt different training settings. A major difference is that [28] train their models using PGD AT while the authors use APGD AT, which as shown in Tab A2 causes a significant robustness boost +1.4% (ConvNeXt-B APGD AT outperforms ConvNeXt-B PGD only by 0.3-0.4%). Besides, results were obtained with a single run also enhances my concern about these marginal improvement. Therefore, I am not convinced about the conclusion that ConvStem generalize to large non-isotropic architectures. Even supposing ConvStem benefits ConvNeXt regardless of size, it is only tested for one particular family of non-isotropic architecture. It's unknow for me if ConvStem can generalize to others. As this is a pure empirical work, it is improtant to test varied architectures for reliability. Replacing ConvStem in InternImage with PatchStem is a way to test so for me.
> >
> > > Generalization to alternative adversarial training methods.
> >
> > Experiments in this work are based on only one type of adversarial training: APGD adversarial training. (one more result of PGD AT is added during rebuttal) APGD AT is not SOTA AT and there are many more advanced AT methods improving over PGD AT significantly regarding either efficiency (like N-FGSM) or effectiveness (like AWP, HAT). It is unclear how ConvStem will perform if it is trained with these alternative adversarial training methods, and if the observed benefit will hold across the varied adversarial training methods.
> >
> > * N-FGSM: Pau de et al., Make Some Noise: Reliable and Efficient Single-Step Adversarial Training, NeurIPS 2022
> > * AWP: Dongxian et al., Adversarial Weight Perturbation Helps Robust Generalization, NeurIPS 2020
> > * HAT: Rahul et al., Reducing Excessive Margin to Achieve a Better Accuracy vs. Robustness Trade-off, ICLR 2022
> >
> > > Missing related works of defense against unseen attacks.
> >
> > training w.r.t. Linf to defend L1/2 is a case of defense against unseen attacks. How is the performance of ConvStem compared to theirs? Will the performance of ConvStem disappear or be enhanced if these techniques are applied jointly?
> >
> > defense against unseen attacks:
> > 1. David et al., Confidence-Calibrated Adversarial Training: Generalizing to Unseen Attacks, ICLR 2020
> > 2. Cassidy et al., Perceptual Adversarial Robustness: Defense Against Unseen Threat Models, ICLR 2021
> > 3. Sihui et al., Formulating Robustness Against Unforeseen Attacks, NeurIPS 2022
> >
> > it may be also worth comparing defense against union of attacks because you both concerns robustness of multiple threat models:
> > 1. Florian et al., Adversarial Training and Robustness for Multiple Perturbations, NeurIPS 2019
> > 2. Pratyush et al., Adversarial Robustness Against the Union of Multiple Perturbation Models, ICML 2020

---

> > > ### Author Response · Authors · 2023-08-16
> > >
> > > Thanks a lot for your reply. We stress that we already provided several additional experiments in the previous response addressing the reviewers’ questions which confirmed the validity of our approach and results. Below we check all requested experiments which again confirm our results and hope that the reviewer can appreciate our efforts.
> > >
> > > Regarding the criticism of a single run, we note that we are an academic lab and the extensive experiments of this paper were the maximum we could do. Given that the improvements are consistent across architectures/sizes, we believe this is sufficient even without multiple runs.
> > >
> > > >Effectiveness of ConvStem for non-isotropic architectures.
> > >
> > > The ConvStem consistently provides significant improvements in terms of generalization of $\ell_1$- and $\ell_2$-robustness for ConvNeXts of all sizes*, and also improves, although sometimes by a small amount, the robustness in $\ell_\infty$ achieving SOTA performance across settings. Considering that it also is very beneficial for isotropic architectures, ConvStem has a positive, or at worst neutral, impact in all tested cases.
> > >
> > > On reviewers request we worked hard to check replacing the ConvStem from the InternImage-T architecture with a PatchStem (PS). Below is a 50 epochs run with 2 attack steps with our AT training scheme.
> > >
> > > |Model|clean|$\ell_\infty$|$\ell_2$|$\ell_1$|
> > > |-|-|-|-|-|
> > > |InternImage-T|72.9|47.0 |48.5|25.9|
> > > |InternImage-T-PS| 72.7|47.0|47.2|24.5|
> > >
> > >
> > > While the gains are smaller than for ConvNeXt-T, the ConvStem again improves generalization to $\ell_1$ (+1.4%) and $\ell_2$ (+1.3%).  We can test larger models for the final version but due to time/compute constraints this is infeasible before the response deadline.
> > > > ConvNeXt-B+ConvStem vs. ConvNeXt-B [28] can be unfair.
> > >
> > > The ConvNeXt-B[28] uses 300 epochs of training with 3-step PGD (55.82% robust acc.), and is outperformed by our ConvNeXt-B+ConvStem trained for 250 epochs with 2-step APGD (56.14% robust acc.). This means 1200 forw./back. passes (per image) for the ConvNeXt-B[28] versus 750 forw./back. passes for our ConvNeXt-B+ConvStem. Thus, we can achieve this high $\ell_\infty$-robustness with significantly less training time, but much better generalization to $\ell_1$- and $\ell_2$. If in any case the comparison is unfair, then to our disadvantage.
> > >
> > > Regarding APGD-2 vs PGD-2 it is non-obvious if one can transfer the improvement of 1.4% from a much smaller ConvNeXt-T and 50 epochs training to a more than 3x larger ConvNeXt-B trained for 250 epochs. We think that a better optimizer for the inner problem matters more for shorter training than for longer training.
> > >
> > > >Combination with other adversarial training schemes
> > >
> > > We test the architectures in several training setups (see Table 1), and the ConvStem provides improvements in all of them. Thus, we expect similar improvements with other AT schemes. For ImageNet prior work of [12] and [28] used both PGD-AT.
> > > Among the methods mentioned by the reviewer, N-FGSM has the goal of preventing catastrophic overfitting, which we did not experience. AWP and HAT have not been used for ImageNet, but only for smaller datasets. AWP has additional comp. cost due to the weight perturbation.
> > > >Works about generalization of robustness.
> > >
> > > [Stutz et al., [ICLR 2020]]() use a detection mechanism for adversarial robustness, which is not comparable to our setup. [Laidlaw et al., [ICLR 2021]]() do adversarial training on LPIPS-bounded perturbations, not $\ell_\infty$. The work of [Dai et al.[NeurIPS 2021]]() might indeed be related, but they do not use ImageNet and don’t achieve SOTA robustness in the seen threat model.
> > >
> > > Multiple-norm robustness (training for several threat models simultaneously) has a different objective (robustness in the union of the threat models).  However, we understand the reviewer’s request in the sense that we should check if also for multiple-norm robustness the ConvNeXt with ConvStem outperforms PatchStem.
> > >
> > > We use SAT, [Madaan et al. 2021](https://arxiv.org/abs/2006.12135), for multiple-norm AT as it is a good compromise of speed of training and achieved robustness. We always did fine-tuning of the ConvNeXt-T models with the  piecewise-schedule of [Croce et al, 2022](https://arxiv.org/abs/2105.12508) with (2,2,6)-attack steps for $\ell_\infty$, $\ell_2$ and $\ell_1$-threat models resp. We ran other training setups with similar conclusion but can’t report them due to space constraints.
> > > |Model|clean|$\ell_\infty$|$\ell_2$|$\ell_1$|union|
> > > |-|-|-|-|-|-|
> > > |ConvNext-T-PS|71.5|41.3|51.2|41.4|34.6|
> > > |ConvNext-T-CvSt|72.5 |42.9|52.0|43.1|36.7|
> > >
> > > **ConvStem outperforms PatchStem not only in the union of the threat models but also in each individual threat model.** All additional experiments confirm again that the ConvStem consistently outperforms Patch-Stem for the ConvNeXt.
> > >
> > > Given that we could address all questions and concerns with a large number of additional experiments, we kindly ask the reviewer to reconsider their score.

---

> > > > ### Comment · Reviewer_qHdB · 2023-08-17
> > > >
> > > > I appreciate the authors' effort of response and extra experiments. The latest response addresses some of my concerns but not all, so I decided to raise my score from 4 to 5. The remaining concerns are
> > > >
> > > > 1. the robustness improvement regarding Linf is marginal or "neutral" in some cases as agreed by the authors. Besides, the argument against "ConvNeXt-B+ConvStem vs. ConvNeXt-B [28] can be unfair." does not convince me as some major factors like inner optimisation methods are not controlled in these two settings. I know one may not simply transfer that 1.4% improvement across different architectures, but I doubt the authors' argument "a better optimizer for the inner problem matters more for shorter training than for longer training". For example, single-step AT has a relatively worse optimizer and is more likely to suffer from catastrophic overfitting when training longer, in other words, longer training setting benefits more from if single-step is increased to multi-step. I think the best way to clear my original concern is to train a ConvNeXt-B using the exactly same training setting as ConvNeXt-B+ConvStem so that ConvStem is the only variable.
> > > > 2. combination with other AT methods. I acknowledge that ConvStem works with APGD in varied training settings, but this does not guarantee it can generalize to other AT training methods. For example, it is fairly reasonable that one may want to use N-FGSM or any other variant of single-step AT to efficiently train robust models on ImageNet esp. when catastrophic overfitting occurs on large epsilon. The question is whether ConvStem also benefits in this situation or ConvStem should be only used with PGD AT.
> > > > 3. those defenses against unseen attack may not be based on Linf PGD AT but you all try to solve the same problem, i.e., robustness against unseen attacks. I see no reason to not compare this work against them as the robustness improvement against unseen attacks is a major contribution of this work. I can't find any comparison about this so I don't know how good the improved robustness against, e.g., unseen L2 achieved by ConvStem is.

---

> > > > > ### Author Response · Authors · 2023-08-18
> > > > >
> > > > > We thank the reviewer for the quick reply and for updating their score. We appreciate the time they have invested in checking our response.
> > > > >
> > > > > > 1. the robustness improvement regarding Linf is marginal or "neutral" in some cases as agreed by the authors. ...
> > > > >
> > > > > We stated that the improvement **can** be marginal for $\ell_\infty$, it is not marginal for the ConvNeXt-T as well as the ViTs, and it is consistently and significantly improving generalization to $\ell_1$ and $\ell_2$ across all architectures and sizes.
> > > > >
> > > > > Regarding the comparison to [28] (ConvNeXt-B) we just want to repeat that we got a better model in $\ell_\infty$ (+0.32%) with our ConvNeXt-B+ConvStem with only 62.5% (750/1200) of the training time and significantly improved generalization to $\ell_2$ (+4.44%) and $\ell_1$ (+2.11%).
> > > > >
> > > > > However, we offer to train for the final version a ConvNeXt-B model for 250 epochs as well (which takes at least 12 days). On the other hand for the ConvNeXt-T models there is a significant difference across seen and unseen models for the longer training of 300 epochs (Table 1), all required extra experiments Intern-Image-T, ConvNeXt-S (50 epochs) showed a significant difference in generalization to $\ell_1$ and $\ell_2$ on adding ConvStem. Thus there is no evidence to believe that this should be different for the B-model and we find it unfair to hold this point against us. &nbsp;
> > > > >
> > > > >
> > > > > >  2. combination with other AT methods. I acknowledge that ConvStem works with APGD in varied training settings. ...
> > > > >
> > > > > As far as we know, no work has experienced Catastrophic overfitting (CO) on ImageNet at 4/255, see also section 5 in [[Andriuschenko & Flammarion NeurIPS 2020](https://proceedings.neurips.cc/paper/2020/file/b8ce47761ed7b3b6f48b583350b7f9e4-Paper.pdf)]. Also [12] used 1-step training and did not report problems with CO.  We tried 1-step training as well and it worked (note that 1-step APGD is equivalent to 1-step PGD with stepsize 2 epsilon). Hence, there is no evidence that 1-step PGD training with ConvStem will experience CO.
> > > > > We also repeat that prior [12] and concurrent [28] work used PGD-AT. One simple reason for this is that most other AT techniques have additional hyperparameters which incurs additional computational cost to fix them. While we see the point that it is interesting to check other AT training techniques for ImageNet, this goes beyond the scope of this paper. And even if hypothetically the improvement of ConvStem were not present for other AT techniques, for which we see absolutely no evidence, then it is sufficient that it works for one training scheme: in fact, in an application only the final robustness and performance counts and not how one has arrived at it.
> > > > >
> > > > >
> > > > >
> > > > > > 3. those defenses against unseen attack may not be based on Linf PGD AT but you all try to solve the same problem. ...
> > > > >
> > > > >
> > > > > Our primary  goal is $\ell_\infty$-robustness and our papers finds that simple architectural changes (PatchStem vs ConvStem) have considerable influence on generalization to the unseen threat models $\ell_1$ and $\ell_2$. We reach SOTA performance for the $\ell_\infty$-threat model. The cited works aim at generalization but they are far from SOTA performance in $\ell_\infty$, e.g. Cassidy Laidlaw et al report in Table 2 that their model has for CIFAR-10, a $\ell_\infty$-robustness of 30.2% at 8/255 whereas they report 49% for an AT model trained for $\ell_\infty$ (which is still far away from SOTA even for the reported model size) and Shihui Dai et al report in Table 2 a robustness at 8/255 of 38.8% (also for CIFAR). None of these works claims that they can achieve SOTA in $\ell_\infty$.
> > > > >
> > > > > In contrast we have shown results for the requested multiple-norm robustness where the reported robust accuracies are also directly comparable to what we reported in the paper and where we see that again ConvStem outperforms PatchStem.
> > > > >
> > > > >
> > > > > While we appreciate the critical comments of the reviewer, we have shown in all additional experiments that the improvements are consistent. We would also like to mention here that pre-trained ImageNet models are used as backbones for training semantic segmentation and object detection models. As stated in a response to reviewer __bR1u__ we could significantly improve robustness for semantic segmentation using our robust ImageNet models as initialization.  Thus we think that this  paper is a valuable contribution which has a direct impact on other applications.

---

### Official Review · Reviewer_bR1u · 2023-07-04

**Soundness:** 3 good
**Presentation:** 3 good
**Contribution:** 3 good
**Rating:** 5
**Confidence:** 4

**Summary:**

The paper has done to some extent comprehensive experimental results to show the effect of ConvStem on improving the robustness of Vit and ConvNeSt model architectures.

**Strengths:**

The experimental results are comprehensive for the proposed purpose.

**Weaknesses:**

There are still some parts that need more clarification;
1- I understand the paper took an experimental point of view to show the effectiveness of the module in improving robustness; however, it would be great to provide more intuition on how and what's the reason for this effect.
2- I would expect a more comprehensive conclusion from the experiments and observations and the author's recommendations for practical applications.
3- Authors mentioned there are "a number of interesting future research directions" but there is no explanation of what are those directions.
4-

**Questions:**

Based on my understanding I still could not answer the questions below:
1- Do ViT architectures are more appropriate for robust or convolutional models?
2- ConvStem had a negative effect on the clean accuracy of ConvNeXT architecture based on Table 2. How can this be explained?
3-

**Limitations:**

As mentioned by the authors some of the well-known architectures have been missed because of computational complexity and this limited the final conclusion from experimental results.

---

> ### Author Rebuttal · Authors · 2023-08-09
>
> We thank the reviewer for their comments and the positive score. We are glad that the reviewer appreciated our extensive experiments. In the following we address the concerns and questions raised in the review.
>
> &nbsp;
>
> **I understand the paper took an experimental point of view to show the effectiveness of the module in improving robustness; however, it would be great to provide more intuition on how and what's the reason for this effect.**
>
> Please refer to the [global response](https://openreview.net/forum?id=Pbpk9jUzAi&noteId=3fBEqU4pBW) above.
>
> &nbsp;
>
> **I would expect a more comprehensive conclusion from the experiments and observations and the author's recommendations for practical applications.**
>
> We tried to present the main observations and takeaways together with the corresponding experiments through the paper. We are happy to expand the comments on these points in the revised version. Regarding practical applications we might recommend using ConvNeXt if only $\ell_\infty$-robustness is required and ViT+ConvStem if generalization across threat models is the goal (please find in the replies below more discussion on these differences).
>
> &nbsp;
>
> **Authors mentioned there are "a number of interesting future research directions" but there is no explanation of what are those directions**
>
> Our observations about the impact of architecture components on adversarial robustness and its generalization might be the starting point for more fine-grained analysis and designing new architectures. Moreover, ImageNet classifiers are often used for transfer learning to other tasks, even outside image classification, e.g. in semantic segmentation or object detection one uses pre-trained ImageNet models as backbones for initialization. First results show that when using robust ImageNet models for initialization of the backbone one can train adversarially robust semantic segmentation models which are significantly more robust than previous work and this even at a lower computational cost.
>
> &nbsp;
>
> **Do ViT architectures are more appropriate for robust or convolutional models?**
>
> While it can be appealing to make strong statements about a single architecture being optimal for adversarial robustness, we think that the matter is more nuanced: from our experiments one observes that ConvNeXts can achieve consistently higher robustness in the $\ell_\infty$-threat model used for training, while ViTs show consistently much better generalization to unseen attacks. One main outcome of our work, besides deriving a training recipe which works across architectures, is to show this more nuanced behavior and how architecture components influence it. Note that it is one of our contributions to check the unseen $\ell_1$ and $\ell_2$-threat models, otherwise the better behavior of ViTs in this regard would have been overlooked.
>
> &nbsp;
>
> **ConvStem had a negative effect on the clean accuracy of ConvNeXT architecture based on Table 2. How can this be explained?**
>
> We note that ConvStem leads to slightly (0.3%) lower clean accuracy only for the base model, while it gives an improvement for ConvNeXt-T. Moreover, it improves robustness in seen and unseen threat models, which is our main metric of interest (please note that if one wants to optimize for clean *and* robust accuracy one should anyway not do pure AT). A possible explanation of this is that the clean ConvNeXt-B+ConvStem used as initialization has 0.8% lower clean accuracy than the clean ConvNeXt-B from the timm library (see Table 9 in App. B.4), since we fine-tuned it for only 50 epochs compared to 300 epochs for the timm model. Thus, the model after the additional 50 epochs of adversarial training might still suffer from this worse initialization.

---

> > ### Author Response · Authors · 2023-08-16
> >
> > Dear reviewer, as the author's response is ending soon, we would appreciate it if you could confirm that our reply has addressed your questions and concerns and if so kindly reconsider your score.

---

### Official Review · Reviewer_zQGv · 2023-07-05

**Soundness:** 3 good
**Presentation:** 3 good
**Contribution:** 3 good
**Rating:** 6
**Confidence:** 3

**Summary:**

This work studies adversarial training on large-scale datasets (i.e., ImageNet), which study the influence of architecture and training schemes on the robustness of classifiers based on different architectures. The paper lacks novelty, but the observations are insightful.
I think this work is a valuable contribution to the research on adversarial training.

**Strengths:**

1  The paper presents a good review of the architectures training strategy and data augmentation, and presents the impact of varying each of them on adversarial robustness and clean accuracy

2 Based on experiments with different settings, the proposed method achieves the sota robustness.



**Weaknesses:**

1. The paper is less novel and more like an experimental analysis report.

2. The other weakness of this paper is that most of the results shown in the paper are quantitative. Lack of some theoretical analysis.


**Questions:**

1 In Table 2, the authors claim the proposed can improve by 5.8% for small, 7.2% for medium, and 2.8% for large models over SOTA.  How to compare and calculate to get this conclusion？ It is unclear.

2  Since Heavy Data Augmentation contains different data augmentation strategies, How do different strategies affect the robustness of the model?

**Limitations:**

Adversarial training on large-scale datasets requires a lot of computing resources, It is not easy to reproduce the experimental results in the paper.

---

> ### Author Rebuttal · Authors · 2023-08-09
>
> We thank the reviewer for the supportive comments and the positive score. We are glad that the reviewer appreciated insights provided by our work and its contributions to the field. In the following we address the weaknesses and questions pointed out in the review.
>
> &nbsp;
>
> **The paper is less novel and more like an experimental analysis report. The other weakness of this paper is that most of the results shown in the paper are quantitative. Lack of some theoretical analysis.**
>
> Please refer to the [global response](https://openreview.net/forum?id=Pbpk9jUzAi&noteId=3fBEqU4pBW) above.
>
> &nbsp;
>
> **In Table 2, the authors claim the proposed can improve by 5.8% for small, 7.2% for medium, and 2.8% for large models over SOTA. How to compare and calculate to get this conclusion？It is unclear.**
>
> For the small models, we compare the ConvNeXt-T of Debenedetti et al. [12] to our best ConvNeXt-T+ConvStem, with robust accuracy wrt $\ell_\infty$ 44.40% and 50.16% respectively, and similarly for medium models we compare the XCiT-M12 [12] to our ConvNeXt-S+ConvStem (45.24% vs 54.42%). For the large models, we do not consider the models from [28] since it is concurrent work (which our classifiers would anyway outperform, see Table 2), and compare the ViT-B from Rebuffi et al. [40] to our ConvNeXt-B+ConvStem (53.50% vs 56.28%). We will clarify this in the revision of the paper.
>
> &nbsp;
>
> **Since Heavy Data Augmentation contains different data augmentation strategies, How do different strategies affect the robustness of the model?**
>
> Exploring which specific augmentation techniques are most helpful for adversarial training would open another large space of expensive (ImageNet-scale) experiments. So, we decided to consider various components of the heavy augmentations as a whole, and switch its components on and off simultaneously, for simplicity.
>
> For a related analysis we conducted, we trained ViT-S + ConvStem with strong clean pre-training, 300 epochs of adversarial training and either basic (random crop and horizontal flip) augmentation or the augmentation scheme of Debenedetti et al. [12], named “3-Aug”, which can be seen as an intermediate step between our basic and heavy augmentations. In Table A1 below (and in the attached file), which includes our model from Table 1, we see that increasing the level of augmentations progressively improves both clean and robust accuracy, and the intermediate augmentation strength is suboptimal. In particular, we observed that lighter augmentation schemes, with the same training protocol used throughout all experiments, lead to overfitting. We will include this discussion in the revised version.
>
> &nbsp;
>
> **Table A1:** effect of different levels of augmentation for ViT-S + ConvStem.
>
> | Model     | Augmentation | clean | $\ell_\infty$ | $\ell_2$ |$\ell_1$ |
> |-----------|:----------:|-------|---------------|----------|---------|
> |ViT-S-CvSt |  basic   | 66.5  |      38.5     |   41.3   |   19.2  |
> |ViT-S-CvSt |  3-Aug   | 70.1  |      43.6     |   46.1   |   23.4  |
> |ViT-S-CvSt |  heavy   | 72.5  |      48.1     |   50.4   |   26.7  |
>
> &nbsp;
>
> **Adversarial training on large-scale datasets requires a lot of computing resources, It is not easy to reproduce the experimental results in the paper.**
>
> Please note that we provided the training scripts in the supplementary material, and we have made the code and the checkpoints of our models already publicly available to foster research in this area.

---

> > ### Comment · Reviewer_zQGv · 2023-08-11
> > **Reply to the author**
> >
> > Thanks for the valuable responses. some of my concerns have been addressed. But there are still some of my concerns unresolved.
> >
> > 1. In Table 2, The baseline is ConvNeXt-T of Debenedetti et al. [12], I think this experimental comparison is unfair. Increasing the training epoch and the iterations of adversarial training can obviously increase the robustness of the model. So the performance gains claimed by the authors are not significant.
> >
> > 2. Heavy Data Augmentation contains different data augmentation strategies, How different strategies affect the robustness is still unknown.
> >
> > If the author can't address these concerns, I will change my score

---

> > > ### Author Response · Authors · 2023-08-11
> > >
> > > Thanks for the quick reply.
> > >
> > > **In Table 2, The baseline is ConvNeXt-T of Debenedetti et al. [12], I think this experimental comparison is unfair. Increasing the training epoch and the iterations of adversarial training can obviously increase the robustness of the model. So the performance gains claimed by the authors are not significant.**
> > >
> > > Please note that we discuss the effect of the number of attack steps in the paragraph _“1-Step vs 2-Step Adversarial Training”_ at the beginning of Sec. 5, with the precise goal of providing a discussion and fair comparison with respect to the training budget.
> > >
> > > Table B1 below summarizes the corresponding results for a detailed comparison to Debenedetti et al. [12]. For each model we additionally show the number of passes of the network (for this, each pass consists of 1 forward + 1 backward pass) for each training image as a proxy of the cost of training (we recall that each attack step requires one pass, and the weights update step requires an additional pass). The ConvNeXt-T of Debenedetti et al. [12] is trained with 1 attack step for 110 epochs (220 passes per image) and achieves 44.4% robust accuracy in $\ell_\infty$. At the same time our ConvNeXt-T+ConvStem with 1 attack step and 50 epochs (100 passes) achieves 46.4%, meaning **an improvement of 2.0% already with less than half the training effort**. ConvNeXt-T+ConvStem with 2 attack steps and 50 epochs (150 passes) further improves robust accuracy to 47.5%, and even ConvNeXt-T (without ConvStem) gets 2.1% better results than the model of [12] even with shorter training (150 passes ours vs 220 of [12]).
> > >
> > > We think that these results clearly show the performance gains brought by our training scheme.
> > >
> > > &nbsp;
> > >
> > > **Table B1:** detailed comparison to [12]. “Passes / image” measures the cost of training.
> > >
> > > | Model     | epochs | attack steps | passes / image| $\ell_\infty$ | improv. over [12] |
> > > |-----------|:----------:|:-------:|:---------------:|:----------:|:---:|
> > > | ConvNeXt-T [12] |  110  | 1 | 220 | 44.4 | — |
> > > | ConvNeXt-T | 50 | 2 | 150 | 46.5 | +2.1 |
> > > | ConvNeXt-T + ConvStem | 50 | 1 | 100 | 46.4 | +2.0 |
> > > | ConvNeXt-T + ConvStem | 50 | 2 | 150 | 47.5 | +3.1 |
> > > | ConvNeXt-T + ConvStem | 300 | 2 | 900 | 49.5 | +5.1 |
> > > | ConvNeXt-T + ConvStem | 300 | 3 | 1200| 50.2 | +5.8 |
> > >
> > > &nbsp;
> > >
> > > **Heavy Data Augmentation contains different data augmentation strategies, How different strategies affect the robustness is still unknown.**
> > >
> > > We first want to remark that studying which configuration is best for adversarial training on ImageNet might constitute a separate project given the large amount of possible research directions and then experiments. As the reviewer mentions, the heavy augmentations we use contain several strategies, and each one has parameters which can be tuned. Moreover, it is not clear the same fine-grained configuration would be optimal for each architecture and model size. Finally, our heavy augmentations simply follow the standard scheme of [30] which is common, with little to no variation, to other works e.g. [29, 40].
> > >
> > > However, we tried to provide some insights on this by testing the lighter augmentation scheme, 3-Aug, from [12] in Table A1 shared in the rebuttal. 3-Aug only uses random flip and crop, and color jitter, and it seems not sufficient to match the results of the heavy augmentations. We additionally provide a similar ablation for ConvNeXt-T+ConvStem in Table B2, with similar observations (we had done this already prior to the rebuttal but forgot to put in our reply).
> > >
> > > &nbsp;
> > >
> > > **Table B2:** effect of different levels of augmentation for ConvNeXt-T + ConvStem (50 epochs).
> > >
> > > | Model     | Augmentation | clean | $\ell_\infty$ | $\ell_2$ |$\ell_1$ |
> > > |-----------|:----------:|-------|---------------|----------|---------|
> > > |ConvNeXt-T + ConvStem |  basic   | 69.1 | 42.2 | 42.5 | 19.7 |
> > > |ConvNeXt-T + ConvStem |  3-Aug   | 71.2 | 45.3 | 45.6 | 19.6 |
> > > |ConvNeXt-T + ConvStem |  heavy   | 69.1 | 47.5 | 47.8 | 24.6 |
> > >
> > > &nbsp;
> > >
> > > As mentioned above, conducting exhaustive ablation experiments is computationally not feasible for us. At the moment our computational resources would allow us to test a very small amount of augmentation configurations for a single architecture until the reply deadline (we are happy to explore this in more detail for the final version). Thus we kindly ask the reviewer to suggest the augmentation configurations they would like to see.

---

> > > > ### Comment · Reviewer_zQGv · 2023-08-20
> > > > **Reply to the author**
> > > >
> > > > Thanks for the authors' responses,  the response addresses my main concern. One suggestion is that as a work mainly based on experiments, it is inappropriate to claim SOTA performance in the paper because most tricks in the experiment have been widely recognized as improving the performance of the model, so I hope the author will weaken this statement in the final version of the paper.
> > > >
> > > > I keep my rating.

---

### Official Review · Reviewer_vpT1 · 2023-07-05

**Soundness:** 3 good
**Presentation:** 4 excellent
**Contribution:** 3 good
**Rating:** 6
**Confidence:** 5

**Summary:**

This work provides extensive experiments to study the influence of architecture and training schemes on the robustness of classifiers to seen and unseen attacks. The paper investigated SOTA model architectures and it variants, and derives a strategy to build better architecture and training scheme to improve model robustness. With the help of all the explored variations, a SOTA robustness on ImageNet is achieved with the newly proposed scheme.

**Strengths:**

This work makes interesting observation on the impacting factors of large-scale adversarial training with extensive experiments. The way the experiments are designed and conducted are solid, and can inspire future work along the direction of systematically understanding the influence of architecture and training scheme on robustness. The observed insights successfully leads to a better architecture and training scheme design that leads to SOTA robustness.

**Weaknesses:**

The main drawback of the work is the lack of theoretical analysis/insight on why some proposed techniques work. Without further insights this work is mainly an extensive ablation study within a design space of existing methods, but cannot infer on the performance for techniques beyond the scope of the emperical explorations. Given the high resource needed to exhaustively explore all architectural ingredients and training scheme design, some theortical insight derived from existing observation will be helpful to guide future exploration.

**Questions:**

See weakness.

**Limitations:**

The limitation of this work is adequately discussed. No potential negative impact is observed.

---

> ### Author Rebuttal · Authors · 2023-08-09
>
> We thank the reviewer for the supportive comments and the positive score. We are glad that the reviewer appreciated the relevance of our work and the SOTA results it achieves. In the following we address the weaknesses pointed out in the review.
>
> &nbsp;
>
> **The main drawback of the work is the lack of theoretical analysis/insight on why some proposed techniques work. Without further insights this work is mainly an extensive ablation study within a design space of existing methods, but cannot infer on the performance for techniques beyond the scope of the emperical explorations. Given the high resource needed to exhaustively explore all architectural ingredients and training scheme design, some theortical insight derived from existing observation will be helpful to guide future exploration.**
>
> Please refer to the [global response](https://openreview.net/forum?id=Pbpk9jUzAi&noteId=3fBEqU4pBW) above.
>
> &nbsp;
>
> We would be happy to discuss any additional points of interest for the reviewer.

---

> ### Comment · Reviewer_vpT1 · 2023-08-21
>
> I would like to thank the author for the detailed response. I will keep my score.

---

### Official Review · Reviewer_LmXJ · 2023-07-07

**Soundness:** 3 good
**Presentation:** 3 good
**Contribution:** 2 fair
**Rating:** 5
**Confidence:** 5

**Summary:**

This paper investigates the impact of architectures and training strategies on adversarial robustness in adversarial training. The substitution of PatchStem with ConvStem leads to an improvement in adversarial robustness, not only against l_inf attacks but also against previously unseen attacks like l_1 and l_2 attacks. Furthermore, the effects of strong pre-training and extensive data augmentation on adversarial robustness have been examined through ablation studies.

**Strengths:**

This paper extends the discussion of adversarial robustness to larger datasets and models, which is significant for the application of deep models. The paper explores one of the most effective methods, adversarial training, from various aspects such as model architecture and training techniques, to improve the adversarial robustness of the models. It proposes replacing PatchStem with ConvStem in the model architecture, resulting in enhanced robustness against l_inf adversarial attacks as well as significant improvements in robustness against unseen attacks, such as l_1 and l_2 attacks. Additionally, the paper validates the positive impact of pre-training and data augmentation in adversarial training. The final results also achieve state-of-the-art performance. The paper is well-structured and easy to understand.

**Weaknesses:**

This paper has a relatively weak innovation aspect, and a significant portion of the experiments leans towards engineering-oriented approaches. Although the results are relatively good, the paper lacks in-depth analysis of the improvement brought about by ConvStem. The reasons behind the improved performance on unseen attacks are still unknown. To further support this claim, it is necessary to explore different attack settings, including black-box attacks.

**Questions:**

It appears that ConvStem performs better on unseen attacks. However, it would be interesting to investigate its performance on black-box attacks as well. One relevant black-box attack is VMI-FGSM (Wang, Xiaosen, and Kun He. "Enhancing the transferability of adversarial attacks through variance tuning." Proceedings of the IEEE/CVF Conference on Computer Vision and Pattern Recognition. 2021). To support your opinion, it would be beneficial to compare the performance of the models listed in Table 2 on some black-box attacks. This analysis would provide a more comprehensive understanding of the models' robustness in the face of different attack scenarios.

---

> ### Author Rebuttal · Authors · 2023-08-09
>
> We thank the reviewer for the detailed comments. We are glad that they appreciated the relevance of our work as well as its experimental results. In the following we individually address the weaknesses pointed out in the review.
>
> &nbsp;
>
> **This paper has a relatively weak innovation aspect, and a significant portion of the experiments leans towards engineering-oriented approaches. Although the results are relatively good, the paper lacks in-depth analysis of the improvement brought about by ConvStem. The reasons behind the improved performance on unseen attacks are still unknown.**
>
> Please refer to the [global response](https://openreview.net/forum?id=Pbpk9jUzAi&noteId=3fBEqU4pBW) above.
>
> &nbsp;
>
> **It appears that ConvStem performs better on unseen attacks. However, it would be interesting to investigate its performance on black-box attacks as well. One relevant black-box attack is VMI-FGSM (Wang, Xiaosen, and Kun He. "Enhancing the transferability of adversarial attacks through variance tuning."... it would be beneficial to compare the performance of the models listed in Table 2 on some black-box attacks. This analysis would provide a more comprehensive understanding of the models' robustness in the face of different attack scenarios.**
>
> We clarify that with unseen attacks we mean **threat models different from the one used during training**, in our case $\ell_\infty$-bounded perturbations (seen) vs $\ell_2$- and $\ell_1$-bounded ones (unseen).
>
> Among the attacks of AutoAttack (AA) (APGD on CE/DLR loss, FAB attack, Square attack) which we use for evaluation, APGD on DLR loss, FAB and Square attack are also “unseen”, in the sense that they are different from the APGD on CE with 2 steps used for training, and even APGD in CE in AA uses 100 steps. However, they all use the same threat model, i.e. $\ell_\infty$-bounded perturbations.
>
> We recall that Square Attack in AA is a **SOTA black-box score-based attack**. In our experiments, Square Attack never found a successful adversarial perturbation for a point where the white-box attacks failed (i.e. Square Attack does not reduce the robust accuracy any further). Transfer black box attacks such as the suggested VMI-FGSM are typically significantly weaker than white box attacks: as we are interested in the actual robustness and not the robustness against a specific attack, transfer attacks are highly unlikely to improve the robustness evaluation. However, we are happy to include such evaluation if the reviewer could clarify why this result would make our experiments more complete and which model we should use for the transfer attack.

---

> > ### Author Response · Authors · 2023-08-16
> >
> > Dear reviewer, as the author's response is ending soon, we would appreciate it if you could confirm that our reply has  addressed your questions and concerns and if so kindly reconsider your score.

---

> > ### Comment · Reviewer_LmXJ · 2023-08-19
> >
> > The rebuttal addresses most of my concerns. I improve the rating.

---

### Author Rebuttal · Authors · 2023-08-09

### General comment

We thank all reviewers for their feedback. We appreciate that all reviewers think that our findings are important and highlight our extensive experiments. We address here the main points shared by multiple reviewers, and reply in detail to specific questions individually below. We look forward to hearing whether the reviewers’ concerns are resolved or they have additional questions.

&nbsp;

### Response to common questions

**Lack of theoretical analysis**

Theoretical analyses of both new architectures and ingredients of adversarial training are very rare in the literature, and empirical findings are those driving the advancements in these areas. In fact, the works, e.g. [Dosovitskiy et al. [2019]](https://arxiv.org/abs/2010.11929), [Liu et al. [2021]](https://arxiv.org/abs/2103.14030), [Liu et al., [2022]](https://arxiv.org/abs/2201.03545), introducing modern architectures do not motivate or study their design with theoretical arguments but rather with extensive empirical explorations and (post-hoc) high level intuitions of what makes them successful. Similarly, [Gowal et al. [2020]](https://arxiv.org/abs/2010.03593), [Pang et al. [2020]](https://arxiv.org/abs/2010.00467), [Rebuffi et al. [2021]](https://openreview.net/forum?id=kgVJBBThdSZ) improved the effectiveness of adversarial training for small datasets (mostly CIFAR-10) by searching for the best training configuration and augmentations. Despite the lack of theoretical insights in those works, they have been of great help for the community and subsequent research, since they advanced the state-of-the-art in adversarial defenses and provided clear indications about which factors are most important for training robust models. We think that our work follows this line of research, and makes an important step towards robust classifiers on ImageNet, which presents specific challenges and opportunities. While we could not theoretically explain the different behaviors across architectures (including the role of the ConvStem), our experiments

1. uncover the existence of such differences in particular regarding generalization to unseen threat models which has not been explored in the literature yet,
2. provide empirical insights about how to successfully train robust classifiers (e.g. clean pre-training, heavy augmentations),
3. show that increasing test-time resolution helps even for robust accuracy.

Some of these indications are even in contrast with those suggested by prior works, and allow us to obtain SOTA $\ell_\infty$-robustness.

&nbsp;

**Analysis of the ConvStem**

We did analyze in detail the design and impact of ConvStem in an ablation study in App. B. In particular, the stem for ViTs (patch size is 16x16) allows for a larger search space since it contains four downsampling layers, compared to the two of a ConvNeXt (patch size 4x4): in Table 8 (App. B.2) one can observe that having only a single downsampling operation per layer yields better robustness than double downsampling or PatchStem (i.e. 16x downsampling with one layer only). This is a possible factor which makes the ConvStem effective, and might also explain the smaller, but positive, impact of ConvStem on the non-isotropic architectures where only two downsampling layers are contained in the stem. In fact, for isotropic architectures all downsampling operations are carried out in the stem (for a total of 16x resolution reduction), while non-isotropic architectures have downsampling layers deeper in the network (only 4x reduction in the stem). Finally, Tables 6, 7 and 8 illustrate the positive effect of the ConvStem on generalization of robustness.

&nbsp;

**Lack of novelty**

We think our work contains several novel aspects: up to our knowledge this is the first work systematically studying the generalization of robustness to unseen threat models on ImageNet. We show that using ConvStem instead of PatchStem yields a large boost in generalization to unseen attacks while still improving in the target threat model. Moreover, we are the first work showing that increasing test-time resolution also boosts robust accuracy which could not be expected as for $\ell_\infty$ increasing resolution yields a more powerful threat model. Note that such differences arise only for high resolution datasets and thus these are new research directions compared to the existing literature on CIFAR. Additionally, while most elements we studied (architectures, pre-training, heavy augmentations) have already been used in some form before, our work empirically analyzes how they interact with each other in the context of adversarial training on ImageNet, which was previously missing. This allows us to distill a new training scheme which effectively improves the robustness of classifiers with various architectures (ViTs, CNNs…), achieving SOTA performance and generalization to unseen threat models.

---

### Author Response · Authors · 2023-08-18

While, the original paper had an extensive number of experiments **validating the efficacy of ConvStem and the proposed training strategy**, on request of the reviewers we performed a large number of additional experiments which confirm and further strengthen our results. As they are scattered over different responses and the PDF of the rebuttal, we have collected all of them here for the convenience of the reviewers and the AC.


**Table: Augmentations.** Heavy data augmentation yields better clean and robust accuracy (seen and unseen threat models) than other weaker augmentation strategies, e.g. 3-Aug suggested in Debenedetti et al [12], validating our use of heavy augmentation.

| Model     | Augment. | clean | $\ell_\infty$ | $\ell_2$ |$\ell_1$ |
|-----------|----------|-------|---------------|----------|---------|
|ConvNeXt-T + ConvStem |  basic   | 69.1 | 42.2 | 42.5 | 19.7 |
|ConvNeXt-T + ConvStem |  3-Aug[12]   | 71.2 | 45.3 | 45.6 | 19.6 |
|ConvNeXt-T + ConvStem |  heavy   | 69.1 | 47.5 | 47.8 | 24.6 |
|ViT-S+ ConvStem |  basic   | 66.5  |  38.5     |   41.3   |   19.2  |
|ViT-S + ConvStem |  3-Aug[12] | 70.1  |  43.6    |   46.1   |   23.4  |
|ViT-S + ConvStem |  heavy   | 72.5  |  48.1    |   50.4   |   26.7  |

**Table: Seemingly unfair comparison to [12] with respect to training time.** Both our ConvNeXt and the ConvNeXt+ConvStem improve over [12] while using less forward/backward passes per image (discussed at the beginning of Section 5.) Total number of forward+backward passes per image combines attack steps with training epochs.

| Model     | epochs | attack steps | passes / image| $\ell_\infty$ |
|-----------|:----------:|:-------:|:---------------:|:----------:|
| ConvNeXt-T [12] |  110  | 1 | 220 | 44.4 |
| ConvNeXt-T | 50 | 2 | 150 | 46.5 |
| ConvNeXt-T + ConvStem | 50 | 1 | 100 | 46.4 |
| ConvNeXt-T + ConvStem | 50 | 2 | 150 | 47.5 |
| ConvNeXt-T + ConvStem | 300 | 2 | 900 | 49.5 |
| ConvNeXt-T + ConvStem | 300 | 3 | 1200| 50.2 |

**Table: Comparison PatchStem vs ConvStem for medium models.** We provide the additional PatchStem models for comparison (all trained for 50 epochs) which confirms
improvements (ViT) or constant (CN) $\ell_\infty$-robustness while having significant gains in terms of the unseen $\ell_2$ and $\ell_1$ threat model for the ConvStem models.

| Model          | clean | $\ell_\infty$ | $\ell_2$ |$\ell_1$ |
|----------------|-------|---------------|----------|---------|
|ViT-M           | 71.7  |      47.2     |   49.0   |   29.2  |
|ViT-M-CvSt      | 72.4  |      48.8     |   50.6   |   28.1  |
|ConvNext-S      | 74.1  |      52.3     |   43.8   |   19.5  |
|ConvNext-S-CvSt | 74.1  |      52.4     |   50.9   |   25.6  |




**Table: Varying test-time resolution** yields also for smaller or larger radii of the test threat model improvements in terms of $\ell_\infty$-robust accuracy of models trained for 4/255  which is a novel finding in our paper. The model was trained at a resolution of 224.

| Perturbation   |  192  |  224  |  256  |  288  |  320  |
|----------------|-------|-------|-------|-------|-------|
|clean           | 74.1  | 75.9  | 76.9  | **77.7**  | 77.2  |
|2/255           | 64.6  | 66.9  | 67.9  | **68.6**  | 68.4  |
|4/255           | 53.0  | 56.1  | **57.3**  | 57.2  | 56.6  |
|6/255           | 41.0  | 43.8  | 44.4  |**44.5**  | 43.0  |
|8/255           | 29.5  | 30.4  | **31.0**  | 29.8  | 27.9  |

**Table: ConvStem for other non-isotropic architectures.** On request we replaced the ConvStem(CS) of the non-isotropic InternImage architecture with a Patchstem (PS). As for the non-isotroic ConvNeXt  generalization to unseen threat models improves for CS compared to PS.
|   	 Model         |  clean | $\ell_\infty$ | $\ell_2$ | $\ell_1$ |
|--------------------------|----------|---------------|-------------| -----------|
| InternImage-T-CS |    **72.9** | 47.0         |    **48.5**     |   **25.9**   |
| InternImage-T-PS |   72.7  | 47.0         |   47.2      |  24.5  |

**Table: Multiple-norm robustness.**  ConvStem (CS) is also better across all threat models over PatchStem (PS) when doing multiple-norm finetuning of the corresponding
$\ell_\infty$-robust models.

|Model|clean|$\ell_\infty$|$\ell_2$|$\ell_1$|union|
|-|-|-|-|-|-|
|ConvNext-T-PS|71.5|41.3|51.2|41.4|34.6|
|ConvNext-T-CvSt|72.5 |42.9|52.0|43.1|36.7|

---

### Decision · Program_Chairs · 2023-09-21

**Decision:**

Accept (poster)

**Comment:**

The paper studies the effect of deep network architecture components on training adversarially robust models on Imagenet. The paper presents a fairly thorough investigation into the effect of different architectural components, as well as transfer robustness. The paper presents some interesting findings. All the reviewers recommend acceptance but raise fair concerns about the practical implications of this study, which the authors are encouraged to address to the best they can in the next version.